# Are Time-Indexed Foundation Models the Future of Time Series Imputation?

**Etienne Le Naour***                                    *etienne.le-naour@edf.fr*
*EDF R&D, Palaiseau, France*

**Tahar Nabil***                                         *tahar.nabil@edf.fr*
*EDF R&D, Palaiseau, France*

**Adrien Petralia**                                      *adrien.petralia@edf.fr*
*EDF R&D, Palaiseau, France*

**Ghislain Agoua**
*EDF R&D, Palaiseau, France*

**Reviewed on OpenReview:** *https://openreview.net/forum?id=cTk56KpsP5*

## Abstract

Foundation models for time series imputation remain largely unexplored. Recently, two such models, `TabPFN-TS` and `MoTM`, have emerged. These models share a common philosophy that places them within the family of time-indexed foundation models. This paper presents the first large-scale empirical study of these models for zero-shot imputation, which enables missing value recovery without retraining across a wide range of scenarios. We conduct extensive univariate experiments across 33 out-of-domain datasets ($\approx$ 1.3M imputation windows) and evaluate their ability to integrate covariates at inference time to improve accuracy without fine-tuning. Our results demonstrate that time-indexed foundation models are a powerful and practical step toward achieving general-purpose, zero-shot imputation for real-world time series. Code is available at `https://github.com/taharnbl/tsfm_imputation`.

## 1 Introduction

Real-world time series from domains such as healthcare, industry, and climate science are often irregularly sampled or incomplete due to sensor failures and decentralized data collection (Schulz & Stattegger, 1997; Clark & Bjørnstad, 2004). Reliable imputation is thus a critical first step toward downstream tasks like forecasting, classification, or anomaly detection. Yet, while recent deep learning methods have advanced imputation performance (Cao et al., 2018; Du et al., 2023a; Nie et al., 2024), they typically lack robustness to distribution shifts and fail to generalize to out-of-domain data.

Recently, zero-shot forecasting models have emerged in the time series community, enabling inference on unseen datasets without retraining. This shift has given rise to time series foundation models, offering key benefits: (i) a single deployable model across diverse domains, (ii) strong performance on new datasets, often exceeding supervised baselines and (iii) emerging capabilities beyond simple memorization. While forecasting-oriented foundation models are now relatively well-studied (Auer et al., 2025b; Das et al., 2024; Woo et al., 2024; Ansari et al., 2024), imputation-focused counterparts remain scarce. Zero-shot imputation in out-of-domain settings is particularly challenging due to heterogeneous sampling rates, diverse missingness patterns, unaligned or irregular time series, and the potential presence of covariates whose predictive value is often underexploited.

---

*Equal contribution.

A promising direction to overcome these challenges lies in continuous-time modeling, which learns a contextual representation $H(t)$ at each time-step $t$, thereby casting imputation as a regression problem. This approach is often referred to as *time-index modelling* (Woo et al., 2023). Within this paradigm, time-indexed foundation models such as `TabPFN-TS` (Hoo et al., 2025) and `MoTM` (Le Naour et al., 2025) have recently been introduced, enabling zero-shot imputation through continuous-time representations. These models offer strong out-of-domain generalization without requiring retraining, making them particularly attractive for real-world applications where training data and computational resources are limited.

In this paper, we conduct the first extensive study of time-indexed foundation models for time series imputation, and summarize our key contributions as follows:

- **Extensive univariate evaluation.** We evaluate `TabPFN-TS` and `MoTM` across 33 out-of-domain univariate datasets (covering roughly 1.3M windows to impute), benchmarking them against a wide range of baselines. `TabPFN-TS` yields the highest overall performance with a notable margin, whereas `MoTM` also surpasses all supervised and local baselines but remains behind `TabPFN-TS`.

- **Evaluating covariate integration without retraining.** On three complex datasets, we show that both foundation models can seamlessly incorporate additional covariates at inference time, drastically improving imputation accuracy without any covariate-specific pretraining.

- **Limitations and discussion.** We analyze the practical constraints of these approaches and identify scenarios where they are most effective. We also discuss potential directions toward more efficient and generalizable foundation models for time series imputation.

## 2 Considered imputation baselines for the benchmark

This section presents the imputation baselines considered in the benchmark, covering a spectrum of approaches from simple statistical heuristics to modern foundation models. In Section 2.1 we present local and supervised models, which, respectively, require dataset-specific training or rely on handcrafted rules; and in Section 2.2 we present time-indexed foundation models, which generalize across datasets in a zero-shot manner without retraining. Further information on the models and their implementations can be found in Section A.

### 2.1 Local imputers and supervised models

**Local Imputers.** Within the benchmark, local imputation methods serve as fundamental baselines due to their simplicity, interpretability, and low computational cost. These approaches estimate missing values using only neighboring observations or straightforward statistical rules. Representative techniques include `Linear Interpolation`, which connects adjacent observations under a constant-rate assumption; `Last Observation Carried Forward (LOCF)`, which propagates the most recent available value; and the `Seasonal Naive` method, which repeats the last observed value of a given periodicity (e.g., daily or weekly). While these methods perform adequately for small and isolated gaps, they generally fail to capture long-term dependencies, seasonal structures, or nonlinear dynamics commonly observed in real-world time series.

**Supervised Models for Imputation.** Supervised models constitute the conventional approach for tackling complex imputation tasks, where models are trained end-to-end on specific datasets and evaluated on held-out test sets. These `task-specific models` — such as `SAITS` (Du et al., 2023a), `BRITS` (Cao et al., 2018), `CSDI` (Tashiro et al., 2021), `TimesNet` (Wu et al., 2023), or `TimeMixer++` (Wang et al., 2025) — are typically deep learning architectures based on recurrent networks, attention mechanisms, or diffusion processes. Their main advantage lies in their ability to capture intricate temporal dependencies and model complex data distributions through direct optimization on the target dataset. However, their reliance on large training datasets often limits their generalization ability in zero-shot or cross-dataset scenarios, requiring retraining on each new task.

## 2.2 Time-Indexed Foundation Models for Imputation

The emergence of foundation models marks a paradigm shift toward general-purpose approaches for time series analysis. These models are characterized by their *zero-shot* capability: rather than being fine-tuned for the imputation task, they directly apply their pre-acquired knowledge to new data.

Models such as `MoTM` (Le Naour et al., 2025) and `TabPFN-TS` (Hoo et al., 2025) differ from conventional foundation models for time series forecasting, which often rely on patch-based attention architectures or extended LSTM variants (e.g., xLSTM in `TiReX` (Auer et al., 2025b)). Patch-based forecasters are trained to predict ground truth values over a given horizon conditioned on sequences of dense fully-observed contexts. However, such models do not handle the diverse missingness patterns of irregular time series inherent to the imputation task. On the other hand, both `MoTM` and `TabPFN-TS` adopt a continuous-time modeling design that naturally generalizes to unobserved timestamps and allows, at inference time, to: (i) handle irregular or unaligned time series, (ii) operate across different sampling rates, (iii) impute arbitrarily missing regions, and (iv) integrate covariates through concatenation with contextual representations.

In essence, these two *time-indexed* models learn a contextual representation $H(t)$ at every timestamp $t$. A regressor $r_\theta(\cdot)$ is then applied to map $H(t)$ to the observed time series value $x(t)$. Yet, despite their conceptual proximity, both models differ substantially in their architectural design. Below we describe both methods.

**`MoTM` (Mixture of TimeFlow Models).** `MoTM` (Le Naour et al., 2025) extends the continuous-time modeling paradigm by leveraging a sophisticated feature extraction mechanism inherited from the TimeFlow architecture (Le Naour et al., 2024). Its core principle is to represent any time series through a pre-trained basis of modulated Implicit Neural Representations (INRs; see also (Li et al., 2025)).

*(i) Representation Learning via Modulated INR Basis.* Specifically, `MoTM` does not learn a single function for the time series but rather a basis of $K$ distinct INRs. Each INR is a small neural network, parameterized by a hypernetwork (Dupont et al., 2022), that maps a continuous time coordinate $t$ to a feature vector. These basis functions are "modulated" in the sense that their parameters are dynamically generated for each new window, allowing them to capture a wide range of temporal patterns (e.g., trends, seasonalities, high-frequency oscillations) without being restricted to predefined frequencies like Fourier features. For any given timestamp $t$, the rich contextual representation $H(t)$ is formed by concatenating the outputs of all $K$ basis INRs evaluated at that time.

*(ii) In-Context Imputation via Local Regression.* The key mechanism of `MoTM` for imputation lies in its local, in-context fitting procedure. Given a time series with missing values, `MoTM` first considers a context window of observed points. It then fits a simple ridge regressor to learn the linear mapping from the high-dimensional representations $H(t)$ of these observed points to their actual values $x(t)$. This local regressor, fitted specifically on the available context, is finally used to predict the values at any missing timestamp $t_{\mathrm{miss}}$ by applying it to the corresponding representation $H(t_{\mathrm{miss}})$.

This framework naturally extends to: ● *Covariates integration with no retraining.* Assuming full observation, additional contextual information available at timestamp $t$ are simply stacked to the target contextual representation $H(t)$. Ridge adaptation proceeds then in the same way as in the univariate setting, leaving the pretrained basis of INRs unchanged. ● *Quantification of uncertainty.* By replacing the ridge regressor with a quantile regressor, `MoTM` can generate quantile predictions to produce uncertainty quantification intervals around the imputed values.

**TabPFN-TS.** Hoo et al. (2025) apply the continuous-time modeling philosophy by reframing the time series imputation task as a standard tabular regression problem. This allows the direct application of the powerful, pre-trained `TabPFN` model (Hollmann et al., 2025) for zero-shot time series analysis. The model's design philosophy is conceptually inverse to that of `MoTM`: it pairs a simple, handcrafted feature representation with a highly expressive regression model.

*(i) Handcrafted Temporal Representations.* Unlike `MoTM`'s learned representations, `TabPFN-TS` employs a straightforward feature engineering approach. The contextual representation $H(t)$ for each timestamp $t$ is constructed by combining the normalized time index itself with a set of pre-defined Fourier basis functions

(i.e., sine and cosine pairs). These Fourier features are chosen to capture key seasonalities expected in the data (e.g., daily, weekly). This method results in a simple, fixed feature set that explicitly encodes temporal position and periodicity, serving as the input for the regression model (see Section A.3.2).

*(ii) Imputation via In-Context Learning with `TabPFN`.* The core expressive power of `TabPFN-TS` resides in its regressor, the `TabPFN` model. `TabPFN` is a large transformer-based architecture pre-trained on hundreds of millions of synthetically generated tabular regression tasks. Its defining characteristic is in-context learning: at inference time, it ingests a set of observed data points—pairs of temporal features and their corresponding values, $(H(t_{obs}), x(t_{obs}))$ as a single "prompt." The model processes this entire context within its attention layers to infer the underlying functional relationship between the features and the series values. It then applies this inferred function to predict the values $x(t_{\text{miss}})$ for the query features $H(t_{\text{miss}})$ of missing timestamps, all within a single forward pass and without any gradient-based fine-tuning.

This framework also naturally supports the *Integration of covariates with no retraining* and *Uncertainty quantification.* In particular, `TabPFN` inherently models uncertainty by returning distribution over outputs.

## 3 Experiments

We design our experimental study to assess two key aspects: (i) zero-shot generalization across out-of-domain datasets and (ii) the ability to incorporate auxiliary covariates without retraining. Thus, experiments are organized into two main parts: a large-scale univariate benchmark covering 33 datasets (Section 3.1), and a focused covariate integration study on three datasets (Section 3.2).

### 3.1 Univariate Benchmark: Out-of-Domain Zero-Shot Imputation

In this section, we evaluate the out-of-domain (OoD) zero-shot performance of `TabPFN-TS` and `MoTM` across 33 diverse real-world datasets. These datasets cover a wide range of sampling rates (5min, 10min, 15min, 30min, 1h) and exhibit heterogeneous temporal patterns and seasonalities. They originate from open-source collections spanning multiple domains, including climate, energy, traffic, etc. Most are drawn from *LOTSA* (Woo et al., 2024) and *GIFT-eval* (Aksu et al., 2024), with strict safeguards to prevent leakage from the three pretraining datasets of `MoTM`. For all details on the datasets please refer to Section B.1. In total, the evaluation involves more than 1.3M incomplete windows.

**Protocol and Baselines.** All datasets are split chronologically intro train, validation and test fractions with respective ratios specified in Section B.1. The test split is divided into four-week segments. For each window, we generate four distinct missing data scenarios, by randomly removing: either (i) 50% and (ii) 70% of the observations (*Pointwise* scenarios); or (iii) two and (iv) four entire days (*Block* scenarios). Note that only the supervised baselines (denoted as *Task Specific Models*) benefit from the train and validation sets. The benchmark includes all the methods described in Section 2. Implementation details, hyperparameters, and method-specific configurations are provided in Section A.

### 3.1.1 Quantitative univariate results

The aggregated results of our univariate benchmark are shown in Figure 1, summarizing the mean normalized MAE across all 33 OoD datasets. Each bar represents the overall imputation error of a model averaged over the four missingness regimes. Models are grouped into three categories reflecting their underlying paradigm: (i) *local* methods, imputing without any learned representation; (ii) *task-specific* models, trained in a supervised manner on each dataset; and (iii) *foundation models* evaluated in a fully zero-shot setting. In addition, Figure 2 presents a critical difference (CD) diagram (Demšar, 2006) that compares models via their average ranks across datasets and missingness scenarios. See also Table 6 in Section C for the full per-dataset results and Section C.4 for additional experiments on datasets with lower sampling rates (daily and weekly).

**Results.** As shown in Figures 1 and 2, three takeaways emerge from the benchmark.

*(i) Time-index foundation models lead the benchmark.* `TabPFN-TS` achieves the lowest mean normalized *MAE*, followed by `MoTM`; both outperform all supervised and local baselines despite being evaluated fully

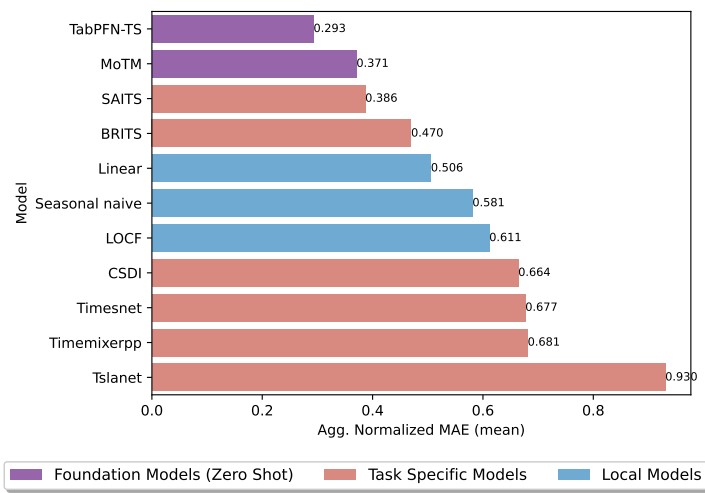

Figure 1: Univariate Benchmark on Out-of-Domain datasets, reported results are z-normalized MAEs.

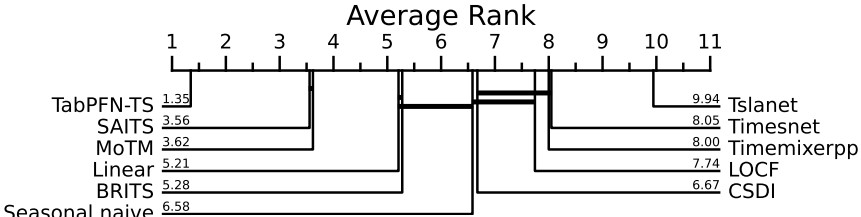

Figure 2: Critical difference diagram over all 33 univariate OOD zero-shot imputation tasks.

zero-shot. In terms of ranking, the CD diagram indicates that `TabPFN-TS` attains the best average rank and is statistically superior to all competitors ($MAE = 0.293$, avg. rank $= 1.35$). While `MoTM` achieves the second-lowest aggregate $MAE$ ($MAE = 0.371$), it ranks third by average rank (avg. rank $= 3.62$) and is not significantly different from `SAITS` under the CD comparison. These results suggest that large-scale synthetic pretrained regressor combined with explicit temporal encodings confers a measurable advantage over the Ridge on top of learned continuous representations.

*(ii) Task-specific models show limited robustness.* While `SAITS` achieves competitive accuracy ($MAE = 0.386$, avg. rank $= 3.56$), other supervised approaches such as `BRITS`, `CSDI`, or `TimesNet` lag behind, sometimes performing worse than simple local heuristics (e.g., `Seasonal Naive`, `LOCF`). This mixed behavior highlights the limited generalization capacity of fully supervised models, which tend to overfit dataset-specific temporal dynamics — particularly when training data is scarce.

*(iii) Local baselines remain resilient.* Classical approaches leveraging temporal priors still deliver reasonable performance in heterogeneous settings, as reflected in both their MAE and ranking consistency. For instance, `Linear Interpolation` achieves an average MAE of 0.506, against e.g 0.664 for `CSDI` or 0.677 for `TimesNet`. However, the pronounced gap separating them from foundation models clearly illustrates the benefits of pretraining time-indexed models, which provide both accuracy and adaptability without retraining.

Overall, the aggregated metrics and the rank-based analysis provide clear evidence that foundation models — particularly `TabPFN-TS` — deliver the most consistent and robust performance for zero-shot time-series imputation across diverse domains. `MoTM` also achieves an honorable level of accuracy despite relying on a comparatively simple regressor, underscoring the effectiveness of its pretrained time-indexed representations. Finally, an ablation study in Appendix D indicates that `TabPFN-TS`'s gains are highly sensitive to the choice of time-index features: the selected handcrafted encoding (normalized time index and daily/weekly Fourier features) consistently outperforms variants that remove periodicities or replace them with random frequencies.

### 3.1.2 Uncertainty quantification results

Beyond pointwise accuracy, it is important to assess how well models capture predictive uncertainty. Both `TabPFN-TS` and `MoTM` natively support quantile estimation, allowing them to produce calibrated uncertainty bounds around each imputed value. Among the baselines, only `CSDI` provides comparable quantile predictions.

We report in Table 1 the average Weighted Quantile Loss (WQL) (see Section C.3 for loss definition) across eleven representative datasets (full results and more details in Section C.3.2). This metric evaluates both imputation accuracy and quantile calibration. The WQL is computed over nine quantile levels, from 0.1 to 0.9, providing a comprehensive measure of the models' probabilistic consistency across the predictions.

Table 1: Weighted Quantile Loss (WQL) average scores on eleven representative univariate datasets.

|  | TabPFN-TS | MoTM | CSDI | SAITS-Q |
|---|---|---|---|---|
| WQL | **0.241** | 0.316 | 0.453 | 0.476 |

**Results.** Quantitatively, `TabPFN-TS` achieves the lowest overall WQL score, followed by `MoTM` and `CSDI`. `SAITS-Q`, a variant of `SAITS` where one model is trained independently per quantile level, obtains the highest WQL. This suggests that foundation-based models such as `TabPFN-TS` and `MoTM` not only reconstruct missing values more accurately but also provide better-calibrated uncertainty estimates.

Qualitative examples in Figure 3 confirm these trends: both foundation models adjust their uncertainty bands (5–95 and 25–75 quantile ranges) to local signal variability. `TabPFN-TS` produces sharper, high-fidelity reconstructions that closely follow the ground truth, while `MoTM` yields smooth yet robust imputations with stable and well-calibrated uncertainty envelopes. Additional examples are provided in Section C.3.3.

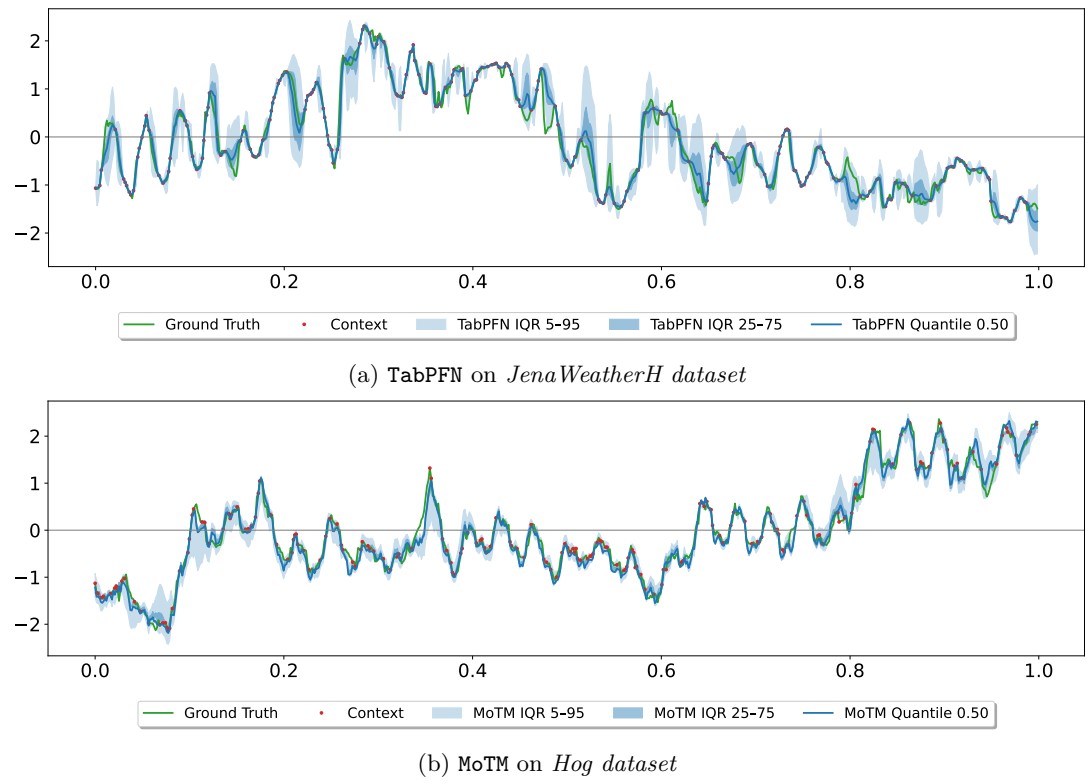

(a) `TabPFN` on *JenaWeatherH dataset*

(b) `MoTM` on *Hog dataset*

Figure 3: Qualitative quantile results in the 70% missing values scenario (*Pointwise 2*).

### 3.2 Integration of Covariates in Zero-Shot OoD Imputation

For a model to be practically deployable in real-world applications, it must be capable of incorporating covariates, which often enhance predictive accuracy by providing additional contextual information. This requirement is recognized as one of the major challenges for foundation models in time series forecasting as well (Auer et al., 2025a). In this section, we investigate how `TabPFN-TS` and `MoTM` integrate additional covariates in a zero-shot imputation setting. We conduct experiments on three datasets where the dependence of the time series values on covariates ranges from weak to critical. This section assumes that the covariate is fully observed; the scenario of partially observed covariates is discussed in Section E.2.

**Datasets, Baselines, and Protocol.**   We evaluate the models on imputation tasks across three datasets: • *PV-France*; aggregated photovoltaic (PV) production curves with the associated global solar irradiance as a covariate, • *Wind-France*; aggregated wind production curves with wind speed as a covariate, and • *Load-France*; aggregated electricity consumption curves with temperature as a covariate. Further details on the datasets and preprocessing are provided in Section B.2.

The train / validation / test splits follow the same protocol as in Section 3.1. We compare the following methods: `TabPFN-TS` and `MoTM` both with or without covariates, a ridge regression using only the covariate to predict the target variable, and `SAITS`. Results are showcased in Table 2.

Table 2: Complete MAE results on datasets with covariates. The best score for each setting is shown in **bold**, second-best is underlined.

| Dataset | Setting | TabPFN-TS | | MoTM | | Other baselines | |
| --- | --- | --- | --- | --- | --- | --- | --- |
| | | TabPFN-TS (W/ Covar) | TabPFN-TS (W/o Covar) | MoTM (W/ Covar) | MoTM (W/o Covar) | Ridge on Covar | SAITS Multivar |
| **PV-France** | *Pointwise 1* | **0.045** | 0.109 | 0.054 | 0.102 | 0.115 | 0.390 |
| | *Pointwise 2* | **0.051** | 0.160 | 0.069 | 0.131 | 0.123 | 0.443 |
| | *Blocks 1* | **0.052** | 0.175 | 0.086 | 0.191 | 0.104 | 0.496 |
| | *Blocks 2* | **0.049** | 0.190 | 0.086 | 0.179 | 0.106 | 0.485 |
| **Wind-France** | *Pointwise 1* | 0.101 | **0.098** | 0.128 | 0.186 | 0.318 | 0.359 |
| | *Pointwise 2* | **0.138** | 0.153 | 0.161 | 0.244 | 0.322 | 0.408 |
| | *Blocks 1* | **0.275** | 0.470 | 0.335 | 0.600 | 0.321 | 0.590 |
| | *Blocks 2* | **0.248** | 0.470 | 0.317 | 0.594 | 0.335 | 0.581 |
| **Load-France** | *Pointwise 1* | **0.037** | 0.037 | 0.138 | 0.146 | 0.667 | 0.292 |
| | *Pointwise 2* | **0.056** | 0.059 | 0.158 | 0.164 | 0.667 | 0.321 |
| | *Blocks 1* | **0.143** | 0.146 | 0.236 | 0.243 | 0.669 | 0.490 |
| | *Blocks 2* | **0.170** | 0.178 | 0.262 | 0.270 | 0.674 | 0.498 |
| **Integrating covariate improvement** | | / | **31.54%** | / | **31.03%** | / | / |

**Results.**   As shown in Table 2, incorporating covariates generally improves performance, both for `TabPFN-TS` and `MoTM`. Substantial gains are observed on datasets for which the covariate strongly informs about the target variable (global irradiance for PV power production, wind speed for wind power production). Under these scenarios and particularly in the challenging block settings, incorporating covariates allows `MoTM` to outperform the impressive univariate performances of `TabPFN-TS`. This emphasizes how fundamental it is to enhance pretrained univariate foundation models with additional contextual information for them to be deployable in real-world applications. On the other hand, weaker relationships, such as between national-level electricity demand and average temperature, yield marginal to no improvement. Overall, `TabPFN-TS (With Covar)` achieves the best results in most scenarios, confirming its strong capacity to integrate heterogeneous contextual inputs for robust zero-shot imputation.

### 3.2.1 Qualitative covariates results

To better visualize the impact of incorporating covariates, we present qualitative results for both `TabPFN-TS` and `MoTM`. Figure 4 and Figure 5 show examples from the *Wind-France* dataset, illustrating how the inclusion of covariate information helps each model reconstruct the four one-day missing blocks more accurately.

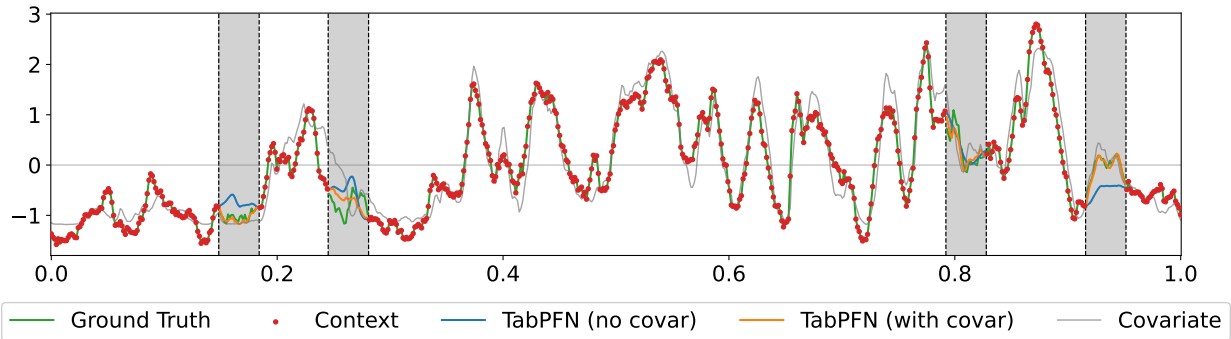

Figure 4: *Wind-France* dataset. `TabPFN-TS` qualitative results with and without covariates in the four one-day missing blocks scenario.

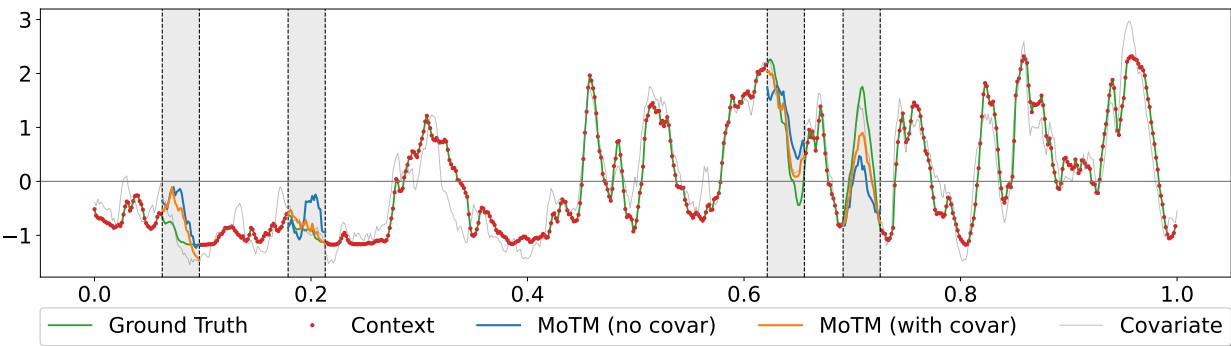

Figure 5: *Wind-France* dataset. `MoTM` qualitative results with and without covariates in the four one-day missing blocks scenario.

**Results.** As illustrated in Figures 4 and 5, incorporating covariates visually improves the reconstructions for both `TabPFN-TS` and `MoTM`. The covariate-enhanced versions capture missing intervals more smoothly and better follow short-term temporal variations within the four one-day gaps. We observe that the interpolations of `TabPFN-TS` align more closely with the ground truth and preserve sharper transitions across missing regions. In contrast, if `MoTM` consistently benefits from covariate information, its reconstructions sometimes struggle to capture sudden changes with great accuracy. More covariate plots are showcased in Section E, confirming that the most visible gains are observed on the *Wind-France* and *PV-France* datasets, while the impact remains limited for *Load-France*.

## 4 Practical considerations

While the previous sections establish the strong zero-shot performance of foundation models, it is equally important to assess their practical limitations and behavioral consistency under different conditions. This section provides complementary analyses covering two key aspects: (i) the computational efficiency of the proposed models across datasets of varying scales, and (ii) their robustness to different missingness patterns.

### 4.1 Computational cost

In this section, we compare the computational efficiency of the proposed models across datasets of varying sizes in Figure 6. The reported times correspond solely to inference for `TabPFN-TS`, `MoTM`, and `Linear`, as these methods operate in a zero-shot manner. For the supervised baseline `SAITS`, we report the total time including both training and inference. The analysis spans four representative datasets — ranging from small (*Covid19 Energy*) to large-scale (*Traffic*) — to illustrate the scalability and practical trade-offs between

methods. For each dataset, the reported MAE results are averaged over the four missing-data scenarios. All experiments are carried out on a single NVIDIA H100-80G GPU.

Figure 6: Compute time (log-scale) versus MAE across datasets of increasing size.

**Results.** As shown in Figure 6, `TabPFN-TS` consistently achieves the lowest MAE but suffers from a substantially higher inference time, especially on large datasets. In contrast, `MoTM` offers a favorable balance between accuracy and efficiency, being up to two orders of magnitude faster while maintaining competitive performance. `SAITS` attains moderate error levels but requires significant computational overhead due to its supervised training phase. Finally, `Linear` is the fastest overall but shows markedly poorer accuracy, particularly on more complex datasets. These results highlight the trade-off between zero-shot generalization and computational cost, emphasizing `MoTM` as a scalable alternative to `TabPFN-TS`.

**Practical deployment considerations.** As stated above, a practical limitation of the best model, `TabPFN-TS`, is its inference cost (see Section 4.1): on an NVIDIA H100 GPU, a forward pass over a 672-step chunk takes approximately one second. This cost can be readily amortized in batch offline imputation or low-frequency pipelines, but may be prohibitive for real-time use cases or deployments operating on thousands of concurrent time series. Accordingly, the clearest deployment recommendation is compute-driven: use `TabPFN-TS` when GPU acceleration is available, and prefer `MoTM` in resource-constrained settings or when throughput across many sequences becomes the primary bottleneck. A more detailed discussion is provided in Appendix D.

## 4.2 Breakdown by missing patterns

A detailed breakdown of results across the four missingness regimes is provided in Section C.2. The analysis highlights a clear dependence of model performance on the structure of the missing values. For sparse pointwise removal (50–70% pointwise removal), `Linear Interpolation` remains highly competitive,

occasionally matching foundation models—indicating that simple local continuity assumptions suffice when gaps are isolated. However, performance rapidly degrades under structured, block-wise missingness (two and four one-day gaps), where both `TabPFN-TS` and `MoTM` maintain stable accuracy and clearly dominate all baselines. Overall, these results confirm the robustness of foundation models to temporally correlated gaps, while emphasizing that local methods remain effective only in low-missingness, unstructured regimes.

## 5 Conclusion and Discussion

Our experiments demonstrate that `TabPFN-TS` achieves very strong zero-shot performance, both in univariate imputation and when integrating covariates. Nevertheless, its inference times remain a significant limitation for some real-world deployment scenarios. `MoTM` also delivers strong zero-shot performance, outperforming all supervised baselines on the univariate benchmark. Similarly to `TabPFN-TS`, its ability at leveraging additional contextual information is remarkable. However, it is generally less accurate than `TabPFN-TS`, although it offers substantially faster inference.

These two highly flexible foundation models represent a significant step forward toward "off-the-shelf" zero-shot imputation solutions applicable across a wide range of domains. A promising avenue to combine performance and efficiency would be to build a more powerful regressor on top of the modulated INR features used by `MoTM`, replacing the current ridge regressor with a model trained via in-context learning, potentially merging the strengths of both approaches.

## Acknowledgements

We would like to thank the authors of all the time series datasets listed in Section B, as well as the authors of the `PyPOTS` Python toolkit (Du et al., 2023b), for enabling this work by their strong contribution to the open-source time series research community. Finally, we thank the reviewers for their thoughtful reviews that helped improve the paper.

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

# A    Baselines details and implementation

In this section, we present the imputers included in our experimental evaluation and detail their implementation. As introduced in Section 2, our selection spans a broad spectrum: from simple naïve baselines, through state-of-the-art supervised deep-learning models, to recent time-indexed foundation models. This diversity ensures coverage of both lightweight, assumption-free methods and advanced architectures capable of capturing long-range and cross-variable dependencies.

## A.1    Local imputers

We considered three naïve baselines: a simple linear interpolation, a last observation carried forward, and a seasonal repeat. Note that each imputer operates independently on each incomplete segment and does not use any cross-variable information.

**Linear.**    This baseline imputes a missing value at timestamp $t$ by linearly interpolating between the closest observed neighbors surrounding the gap. Concretely, it uses the last observation before $t$ and the first observation after $t$ as anchors and fills the interior points on the straight line between them. If a leading gap has no past anchor, we adopt a next-observation-carried-backward (NOCB) fallback; if a trailing gap has no future anchor, we fall back to last-observation-carried-forward (LOCF).

**LOCF.**    This baseline imputes a missing value at timestamp $t$ by copying the most recent available past value (the last observation before $t$). If the series begins with a gap and no past value exists, we perform a single NOCB initialization by copying the first available future observation backward.

**Seasonal Naive.**    This baseline imputes a missing value at timestamp $t$ by using the observation from the previous seasonal period, i.e., the value at $t - S$. The seasonal period $S$ is pre-defined for each dataset based on its dominant frequency (e.g., daily or weekly). If the value at $t - S$ is also missing, the method sequentially searches for an available observation at other seasonal timestamps (e.g., $t + S$, then $t - 2S$, etc.). If this search fails to yield a value, the method falls back to a simple LOCF imputation.

## A.2    Supervised imputers

We considered six supervised deep-learning baselines spanning complementary neural paradigms: recurrent, Transformers, and diffusion-based approaches. These include methods built specifically for imputation as well as recent multi-task time-series backbones (convolutional and token-mixing/Transformer hybrids) that tackle imputation via masked-reconstruction training. This diversity probes different inductive biases and offers a balanced accuracy/efficiency trade-off.

All models implementations are taken from the `PyPOTS` (Du et al., 2023b) Python toolbox and trained on fixed-length windows where a subset of *observed* entries is randomly masked; each model reconstructs these masks from the remaining context (values + binary observation mask). Inputs are z-score normalized per variable, training minimizes the mean absolute error (MAE) on masked positions only, model selection uses validation MSE with early stopping, and test metrics are computed on the 4 missing points scenarios detailed in Section 3.1. For all baselines, we use the default hyperparameter settings recommended by the original authors in their papers or public implementations. In addition, for the strongest supervised baseline, `SAITS`, we perform an extensive hyperparameter study on 11 representative datasets (see Section F). These experiments show that, while careful tuning can yield modest improvements, the default hyperparameter configuration already provides competitive performance at a substantially lower computational cost than tuning each model separately on every dataset. Note that we use the Adam optimizer (Kingma & Ba, 2015) for all models with a batch size of 64, train for at most 50 epochs, and apply early stopping with a patience of 5 epochs.

**SAITS.**    A Transformer imputer with two diagonally-masked self-attention (DMSA) blocks that jointly capture temporal and cross-feature dependencies; a learned gating mechanism combines both blocks to predict missing values efficiently (Du et al., 2023a).

*Hyperparameters.* We use $n_{\text{layers}} = 2$, $d_{\text{model}} = 256$, $n_{\text{heads}} = 4$ with $d_k = d_v = 64$, feed-forward size $d_{\text{ffn}} = 128$ and dropout $= 0.1$. This setting follows the authors' recommended defaults and balances capacity with training stability.

**BRITS.** A bidirectional RNN imputer with learned temporal decay that processes each window forward and backward, jointly estimating hidden states and missing values while enforcing consistency between directions (Cao et al., 2018).

*Hyperparameters.* A single-stack bidirectional GRU with rnn_hidden_size $= 64$ and dropout $= 0.3$. This mirrors common BRITS configurations and provides a compact baseline for irregular gaps.

**CSDI.** A conditional diffusion imputer that models $p(\mathbf{x}_{\text{miss}} \mid \mathbf{x}_{\text{obs}})$ via score-based denoising; training adds Gaussian noise to targets and conditions the denoiser (Transformer/U-Net with time embeddings) on values and masks, enabling stochastic imputations at inference (Tashiro et al., 2021).

*Hyperparameters.* Transformer/U-Net backbone with $n_{\text{layers}} = 4$, $n_{\text{heads}} = 4$, $n_{\text{channels}} = 64$; time, feature, and diffusion embeddings of sizes 128, 32, and 128, respectively. We use $n_{\text{diffusion\_steps}} = 50$, and a quadratic noise schedule type. These values track the public defaults for efficient training while preserving sampling quality.

**TimesNet.** A multi-periodic convolutional model that folds each 1D series into 2D tensors along discovered periods and applies multi-scale 2D CNN blocks to capture intra- and cross-period structure; we train it for masked-value reconstruction (Wu et al., 2023).

*Hyperparameters.* We set $n_{\text{layers}} = 2$, $d_{\text{model}} = 128$, $d_{\text{ffn}} = 256$, $n_{\text{kernels}} = 6$, top_k$= 3$, and dropout $= 0.1$. This lightweight configuration keeps computation modest while retaining the multi-periodic 2D convolutional capacity.

**TimeMixer++.** A token-mixing hybrid (Transformer/MLP) designed for multi-periodic patterns with lightweight mixing blocks and skip connections; adapted here to imputation by reconstructing randomly masked entries from the observed context (Wang et al., 2025).

*Hyperparameters.* We use $n_{\text{layers}} = 3$, $d_{\text{model}} = 64$, $d_{\text{ffn}} = 128$, top_k$= 8$, $n_{\text{kernels}} = 6$, $n_{\text{heads}} = 4$, with channel_mixing and channel_independence, dropout $= 0.1$, and downsampling ($n_{\text{downsampling\_layers}} = 3$ and $n_{\text{downsampling\_window}} = 2$). This mirrors the authors' small/efficient setting adapted for masked reconstruction.

**TSLANet.** A lightweight convolutional model replacing self-attention with an Adaptive Spectral Block (Fourier-domain features with adaptive thresholding) and an Interactive Convolution Block for local–global mixing; trained for single-pass reconstruction of masked entries (Eldele et al., 2024).

*Hyperparameters.* Three layers (n_layers $= 3$) with patch_size $= 16$, embedding dimension $d_{\text{embedding}} = 256$, mask_ratio $= 0.4$, and dropout $= 0.1$ ; matching the default hyperparameters of the official codebase.

### A.3 Time-Index Foundation models

This section complements Section 2.2 by providing further implementation details for the two time-index foundation models evaluated in our experiments: `MoTM` and `TabPFN-TS`.

### A.3.1 MoTM.

**Implementation Details.** Our implementation strictly adheres to the architecture and methodology described in the original paper (Le Naour et al., 2025). We utilize the three publicly available pre-trained TimeFlow (modulated INRs), which were respectively trained on the *Electricity*, *Solar10T*, and *Spanish Temperatures* datasets (further details on these datasets are available in Section B.1). For full reproducibility, both the source code and the pre-trained INR weights were obtained from the official repository: `https://github.com/EtienneLnr/MoTM`.

### A.3.2 TabPFN-TS.

**Implementation Details.**  For our experiments, we utilize the official `TabPFNv2` implementation from the `tabpfn` Python package, which includes the pre-trained `TabPFNv2` model described in (Hollmann et al., 2025). We employ a consistent feature representation, $H(t)$, across all datasets, defined as:

$$H(t) = \left( t, \sin\left(\frac{2\pi t}{P_{\text{day}}}\right), \cos\left(\frac{2\pi t}{P_{\text{day}}}\right), \sin\left(\frac{2\pi t}{P_{\text{week}}}\right), \cos\left(\frac{2\pi t}{P_{\text{week}}}\right) \right)$$

In this formulation, $t$ is the normalized time index, $P_{\text{day}}$ represents the number of timesteps (normalized) in a 24-hour period (e.g., 24 for hourly data, 48 for 30-minute data, 96 for 15-minute data etc.), and $P_{\text{week}}$ is set to $7 \times P_{\text{day}}$ to capture weekly seasonality. For full reproducibility, the source code and pre-trained regressor are publicly available at `https://github.com/PriorLabs/TabPFN/`.

### A.4 Excluded baselines

Our main univariate benchmark includes two foundation models, six deep supervised imputers and three local baselines. Although extensive, this selection could be complemented by several other models. Below we motivate why other relevant foundation models (`MOMENT` and `NuwaTS`), deep supervised models and local baselines (`Spline Interpolation`) where not included in the main experiments. For the sake of completeness, evaluation of these excluded baselines on a representative subset of the benchmark is carried out in Section A.4.1, confirming that they lack robustness to perform well on diverse datasets and settings.

**MOMENT.**  We considered including `MOMENT` (Goswami et al., 2024), a large Transformer-based foundation model pre-trained on patched time series via a masked modeling objective. Initially, our intent was to evaluate it as a zero-shot imputation baseline alongside `TabPFN-TS` and `MoTM`. However, its performance in our benchmark was found to be substantially lower than that of the other methods. A closer review of the original paper clarifies this result: the authors primarily advocate for fine-tuning the model on specific downstream tasks to achieve optimal performance. Moreover, their reported imputation experiments are limited to very short missing segments, a scenario that differs significantly from those addressed in our work. Given its unsuitability for zero-shot time series imputation, we excluded it from our final comparative analysis.

**NuwaTS.**  `NuwaTS` is another Transformer-based model specifically designed to address zero-shot time series imputation by repurposing pretrained language models (Cheng et al., 2024). Although the public codebase[1] does not provide any off-the-shelf pretrained models, the official demonstration[2] allows experimenting with a version pretrained on fixed segments of length 96. To integrate `NuwaTS` into our evaluation pipeline, we adapted this demonstration model by splitting each considered time series into multiple 96-length segments from which `NuwaTS` could extract shared information. This workaround is clearly non-ideal but enables a preliminary zero-shot assessment of the model under comparable conditions. Consistently with the rest of our analysis, these experiments indicate that `NuwaTS` trails substantially behind time-indexed foundation models.

**Other deep learning supervised imputers.**  The field of deep learning for supervised time series imputation is a rapidly evolving area of research. Our selection of baseline models aims to provide a representative sample of this landscape, including both supervised methods that have become standard benchmarks and more recent, state-of-the-art architectures. To ensure fair and reproducible comparisons, all selected deep learning supervised models are part of the `PyPOTS` library (Du et al., 2023b). This framework was essential for conducting our extensive benchmark.

**Spline interpolation.**  `Cubic Spline interpolation` is a simple non-linear imputation method based on piecewise polynomials of third-order. Although it produces smooth interpolations and an accurate fit, this method is known to be prone to overfitting under sparse or noisy observations.

---

[1]Code available at `https://github.com/Chengyui/NuwaTS`.
[2]Demo available at `https://colab.research.google.com/drive/1jjM6g4N7AqyHjYawZWJdbFgNY7p4ZtGY?usp=sharing`.

### A.4.1 Additional experiments

**Setting.** We conduct supplementary experiments to evaluate `MOMENT` (Goswami et al., 2024), `NuwaTS` (Cheng et al., 2024) and a `Cubic Spline Interpolation`. `MOMENT` and `NuwaTS` operate in a fully zero-shot setting. They are compared against `TabPFN-TS`, `MoTM` and `Linear Interpolation`. Table 3 shows their performance on eleven representative univariate datasets (see Section B) and four missingness scenarios (see Section 3.1).

Table 3: Complete MAE results of additional baselines `MOMENT`, `NuwaTS` and `CubicSpline` on 11 datasets. For each setting, the best score (lowest) is in **bold** and the second best is underlined.

| Dataset | Setting | TabPFN-TS | MoTM | MOMENT | NuwaTS | CubicSpline | Linear |
|---|---|---|---|---|---|---|---|
| BDG2–Bear | *Pointwise 1* | **0.171** | 0.202 | 1.026 | 0.349 | 0.281 | 0.229 |
| | *Pointwise 2* | **0.223** | 0.240 | 1.026 | 0.436 | 0.447 | 0.330 |
| | *Blocks 1* | **0.272** | 0.332 | 1.075 | 0.519 | 3.033 | 0.857 |
| | *Blocks 2* | **0.280** | 0.336 | 1.077 | 0.535 | 3.000 | 0.861 |
| BDG2–Rat | *Pointwise 1* | **0.196** | 0.231 | 0.811 | 0.352 | 0.317 | 0.247 |
| | *Pointwise 2* | **0.256** | 0.273 | 0.814 | 0.428 | 0.504 | 0.334 |
| | *Blocks 1* | **0.349** | 0.400 | 0.808 | 0.516 | 4.099 | 0.743 |
| | *Blocks 2* | **0.355** | 0.402 | 0.808 | 0.524 | 4.085 | 0.744 |
| Covid19 Energy | *Pointwise 1* | **0.075** | 0.099 | 1.024 | 0.287 | 0.124 | 0.163 |
| | *Pointwise 2* | 0.132 | **0.127** | 1.021 | 0.381 | 0.272 | 0.292 |
| | *Blocks 1* | **0.202** | 0.222 | 1.075 | 0.457 | 1.689 | 0.949 |
| | *Blocks 2* | **0.201** | 0.232 | 1.073 | 0.474 | 1.107 | 0.928 |
| GFC12 Load | *Pointwise 1* | **0.143** | 0.189 | 0.985 | 0.348 | 0.202 | 0.232 |
| | *Pointwise 2* | 0.237 | **0.231** | 0.985 | 0.436 | 0.419 | 0.370 |
| | *Blocks 1* | **0.353** | 0.382 | 1.043 | 0.486 | 2.622 | 0.810 |
| | *Blocks 2* | **0.363** | 0.387 | 1.047 | 0.499 | 2.682 | 0.814 |
| Hog | *Pointwise 1* | **0.196** | 0.240 | 0.979 | 0.320 | 0.287 | 0.216 |
| | *Pointwise 2* | **0.260** | 0.286 | 0.979 | 0.375 | 0.426 | 0.280 |
| | *Blocks 1* | **0.396** | 0.458 | 1.029 | 0.491 | 3.043 | 0.555 |
| | *Blocks 2* | **0.406** | 0.464 | 1.029 | 0.502 | 2.848 | 0.564 |
| Jena Weather 10T | *Pointwise 1* | 0.086 | 0.190 | 0.982 | 0.143 | 0.101 | **0.080** |
| | *Pointwise 2* | 0.101 | 0.207 | 0.981 | 0.165 | 0.132 | **0.096** |
| | *Blocks 1* | **0.378** | 0.453 | 1.030 | *Nan* | 4.774 | 0.527 |
| | *Blocks 2* | **0.384** | 0.459 | 1.024 | *Nan* | 4.246 | 0.544 |
| Jena Weather 1H | *Pointwise 1* | **0.156** | 0.224 | 0.990 | 0.300 | 0.194 | 0.170 |
| | *Pointwise 2* | **0.214** | 0.272 | 0.983 | 0.364 | 0.343 | 0.243 |
| | *Blocks 1* | **0.366** | 0.463 | 1.078 | 0.475 | 2.698 | 0.552 |
| | *Blocks 2* | **0.370** | 0.464 | 1.055 | 0.489 | 2.605 | 0.555 |
| Oikolab Weather | *Pointwise 1* | **0.150** | 0.227 | 0.984 | 0.318 | 0.176 | 0.164 |
| | *Pointwise 2* | **0.228** | 0.278 | 0.984 | 0.391 | 0.316 | 0.248 |
| | *Blocks 1* | **0.449** | 0.529 | 1.043 | 0.556 | 1.732 | 0.638 |
| | *Blocks 2* | **0.454** | 0.534 | 1.043 | 0.562 | 1.801 | 0.636 |
| PDB | *Pointwise 1* | **0.062** | 0.094 | 0.984 | 0.269 | 0.120 | 0.191 |
| | *Pointwise 2* | **0.119** | 0.121 | 0.977 | 0.339 | 0.278 | 0.338 |
| | *Blocks 1* | **0.171** | 0.197 | 1.051 | 0.352 | 1.200 | 0.982 |
| | *Blocks 2* | **0.177** | 0.201 | 1.053 | 0.382 | 1.801 | 1.006 |
| Pedestrian Counts | *Pointwise 1* | **0.150** | 0.196 | 1.015 | 0.433 | 0.404 | 0.329 |
| | *Pointwise 2* | **0.200** | 0.239 | 1.015 | 0.502 | 0.669 | 0.447 |
| | *Blocks 1* | **0.172** | 0.254 | 1.070 | 0.498 | 3.644 | 1.001 |
| | *Blocks 2* | **0.178** | 0.260 | 1.069 | 0.513 | 5.382 | 1.000 |
| Weather | *Pointwise 1* | **0.257** | 0.298 | 0.966 | 0.360 | 0.359 | 0.259 |
| | *Pointwise 2* | **0.311** | 0.342 | 0.965 | 0.411 | 0.521 | 0.322 |
| | *Blocks 1* | **0.456** | 0.494 | 1.027 | 0.527 | 2.992 | 0.620 |
| | *Blocks 2* | **0.455** | 0.488 | 1.018 | 0.528 | 2.998 | 0.618 |
| **Mean (Std)** | | 0.252 (0.114) | 0.300 (0.122) | 1.002 (0.070) | 0.419 (0.101) | 1.613 (1.535) | 0.502 (0.286) |

**Results.** Table 3 shows the z-normalized MAE scores across all eleven datasets and four missingness settings, enlighting why `MOMENT`, `NuwaTS` and `Cubic Spline Interpolation` where excluded from the main benchmark. (i) `MOMENT` fails to produce accurate predictions across all datasets and settings, confirming that it is not a suitable zero-shot time-series imputation model. (ii) `NuwaTS` outperforms `MOMENT` yet lags significantly behind `TabPFN-TS` on all datasets and settings and behind `MoTM` on ten out of the eleven datasets. Altogether, these observations strengthen the time-indexed approaches as strong alternatives to Transformer-based foundation models for time series imputation. (iii) Finally, the `Cubic Spline Interpolation` suffers from severe overfitting in challenging scenarios, such as high missingness rate (*Pointwise 2*, 70% of missing values) or

block scenarios. By contrast, the `Linear Interpolation` provides more robust predictions and outperforms `Cubic Splines` in 39 out of 44 settings.

# B  Datasets details

## B.1  Univariate datasets

The complete list of datasets used in our univariate experiments is shown in Table 4. 36 datasets were used in total: 3 for the pretraining of `MoTM` and the remaining 33 for the zero-shot evaluation of both `TabPFN-TS` and `MoTM`. Each dataset is split chronologically intro train, validation and test. Unless otherwise stated, the respective fractions are 0.7, 0.1 and 0.2. The test segments were then generated by applying a four-week sliding window, where at every step a random stride is drawn uniformly between 0.5 and 2 days. This procedure ensures that the inference samples are not aligned on any specific calendar information. A short description of each dataset is provided below. Those datasets marked with an `"*"` were curated to remove *flat* segments at inference to avoid biasing the evaluation towards trivial scenarios. Flat segments are caused e.g. by heterogeneous sensor operating dates within datasets or by filling long missing blocks with zeros (Emami et al., 2023).

**Energy domain**

**BDG2-Bear\***, **BDG2-Rat\*** and **Hog\*** are the energy demands of commercial buildings in the US in 2016 - 2017. Sourced by the BuildingsBench library (Emami et al., 2023) from the Building Data Genome 2 (BDG2) project (Miller et al., 2020).

**Borealis\*** and **Ideal\*** contain the total electricity consumption of, respectively, 15 homes in Waterloo, Ontario, in 2011-2012 and 217 homes in Edinburgh, UK, between 2016-2018. Both datasets were released as part of the BuildingsBench dataset, and include a marginal amount of data preprocessing (including interpolation of missing values and outlier removal) (Emami et al., 2023).

**Covid19 Energy** is the aggregated electricity demand of an entire metropolitan area, from 2017 to 2021 (Farrokhabadi et al., 2022).

**GFC12 Load** is sourced from the *Global Energy Forecasting Competition 2012* and contains a total of 20 aggregated load series (Wang et al., 2023).

**PDB** is a Kaggle dataset containing electricity demand and outdoor temperature data in 2013-2014. We omitted the temperature and kept only the electricity demand (Wang et al., 2023).

**Electricity** contains the hourly-aggregated electricity consumption of 370 households in Portugal in 2011-2014 (Trindade, 2015).

**KDD Cup 2022** is a dataset for the Spatial Dynamic Wind Power Forecasting Challenge hosted at KDD in 2022 (Zhou et al., 2022). It contains the wind power data of 134 wind turbines from a wind farm over half a year. We kept the wind power generation variable as the target variable and omitted extra covariates (wind speed and direction, temperature, etc.). The train / validation / test split is 0.65 / 0.15 / 0.2.

**Solar** contains the *synthetic* power production of 137 photovoltaic power plants in Alabama in 2006. Dataset sourced by Lai et al. (2018) using simulations from NREL's Solar Power Data for Integration Studies.

**ETT1** and **ETT2** respectively contain measurements of oil temperature of two electrical transformers in China, as well as six additional covariates. We used these 7 variables in our experiments, handling them with channel independence. The datasets were collected and published by Zhou et al. (2021).

**London Smart Meters Small** is the half-hourly energy consumption of 5561 households in the UK between 2011 and 2014. Data sourced by Godahewa et al. (2020) from `https://data.london.gov.uk/dataset/smartmeter-energy-use-data-in-london-households`. We kept a random subset of 500 samples in our experiments.

Table 4: All datasets used in our univariate experiments and their key properties.

| Dataset | Release Platform | Domain | MoTM Use | Freq | Num. Series | Series Length | Num. Test Segments |
|---|---|---|---|---|---|---|---|
| Electricity | Zenodo | Energy | Train | 1H | 370 | 35064 | 122623 |
| Solar-10T | Zenodo | Energy | Train | 10min | 137 | 52560 | 9179 |
| Spanish Temperatures | Kaggle | Climate | Train | 1H | 5 | 35000 | 1090 |
| BDG2-Bear | LOTSA | Energy | Test | 1H | 91 | 17544 | 7522 |
| BDG2-Rat | LOTSA | Energy | Test | 1H | 280 | 17544 | 24915 |
| Borealis | LOTSA | Energy | Test | 1H | 15 | 7447 | 77 |
| Covid19 Energy | LOTSA | Energy | Test | 1H | 1 | 31912 | 195 |
| GFC12 Load | LOTSA | Energy | Test | 1H | 20 | 39414 | 4960 |
| Hog | LOTSA | Energy | Test | 1H | 24 | 17544 | 2310 |
| Ideal | LOTSA | Energy | Test | 1H | 217 | 16167 | 156 |
| PDB | LOTSA | Energy | Test | 1H | 1 | 17520 | 96 |
| KDD Cup2022 | LOTSA | Energy | Test | 10min | 134 | 35279 | 2546 |
| ERA5 geopotential | LOTSA | Climate | Test | 1H | 500 | 8736 | 19000 |
| ERA5 humidity | LOTSA | Climate | Test | 1H | 500 | 8736 | 19000 |
| ERA5 temperature | LOTSA | Climate | Test | 1H | 500 | 8736 | 19000 |
| ERA5 wind speed | LOTSA | Climate | Test | 1H | 500 | 8736 | 19000 |
| Oikolab Weather | LOTSA | Climate | Test | 1H | 8 | 100057 | 5288 |
| Pedestrian Counts | LOTSA | Transport | Test | 1H | 66 | 96400 | 7733 |
| Traffic | LOTSA | Transport | Test | 1H | 861 | 17544 | 83479 |
| PEMS BAY | LOTSA | Transport | Test | 5min | 325 | 52128 | 2275 |
| PEMS 03 | LOTSA | Transport | Test | 5min | 358 | 26208 | 358 |
| SHMETRO | LOTSA | Transport | Test | 15min | 576 | 8809 | 576 |
| ETT1-15T | GIFT-eval | Energy | Test | 15min | 7 | 69680 | 1050 |
| ETT1-1H | GIFT-eval | Energy | Test | 1H | 7 | 17420 | 1092 |
| ETT2-15T | GIFT-eval | Energy | Test | 15min | 7 | 69680 | 1050 |
| ETT2-1H | GIFT-eval | Energy | Test | 1H | 7 | 17420 | 1092 |
| Solar-1H | GIFT-eval | Energy | Test | 1H | 137 | 8760 | 8768 |
| Jena Weather 10T | GIFT-eval | Climate | Test | 10min | 21 | 52704 | 1428 |
| Jena Weather 1H | GIFT-eval | Climate | Test | 1H | 21 | 8784 | 1344 |
| Loop Seattle 5T | GIFT-eval | Transport | Test | 5min | 323 | 105120 | 21964 |
| Loop Seattle 1H | GIFT-eval | Transport | Test | 1H | 323 | 8760 | 20672 |
| MDense | GIFT-eval | Transport | Test | 1H | 30 | 17520 | 4710 |
| Enedis LDM Small | Zenodo | Energy | Test | 30min | 500 | 17424 | 20500 |
| London Smart Meters Small | Chronos | Energy | Test | 30min | 500 | 22000 | 25779 |
| Spanish Energy | Kaggle | Energy | Test | 1H | 9 | 35064 | 1962 |
| Weather | Informer | Climate | Test | 1H | 11 | 35064 | 2398 |

**Enedis LDM Small** is a dataset of 10k one-year individual electricity consumptions generated by a latent diffusion model at a 30-min sampling rate and representative of thermo-sensitive French households (Nabil et al., 2025). We kept a random subset of 500 samples in our experiments.

**Spanish Energy** is a Kaggle dataset containing the electricity (i) consumption and (ii) production for Spain from 2015 to 2018. We used the total load demand as well as electricity generation of eight energy sources (biomass, fossil gas, fossil hard coal, solar, onshore wind and three technologies of hydropower). We kept these nine variables in our experiments, handling them with channel independence. Data were obtained from https://www.kaggle.com/datasets/nicholasjhana/energy-consumption-generation-prices-and-weather.

**Climate domain**

**Spanish Temperatures** contains the hourly temperature measurements for the five largest cities in Spain, from 2015 to 2018. Data were obtained from `https://www.kaggle.com/datasets/nicholasjhana/energy-consumption-generation-prices-and-weather`.

**ERA5** is part of the ClimateLearn library, which provides historical worldwide time series of various climate (atmosphere and land-surface) variables, including geopotential, humidity, temperature, and wind speed (Nguyen et al., 2023). The dataset extracted by LOTSA is based on a $64 \times 128$ grid structure (Woo et al., 2024). In our experiments, we used data for year 2000 and kept a random subset of 500 samples out of the 8192 available grid points.

**Oikolab Weather** contains hourly measurements of eight meteorological variables from a weather station located near Monash University, Australia (Godahewa et al., 2021). All eight channels are kept in our experiments, treating them as univariate samples (channel independence).

**Jena Weather** contains 21 meteorological indicators, such as air temperature, humidity, etc. collected in 2020 at a 10-minute sampling rate from a weather station in Germany. Sourced by Wu et al. (2021) from `https://www.bgc-jena.mpg.de/wetter/`. All 21 variables are kept in our experiments, treating them as univariate samples (channel independence).

**Weather** contains hourly measurements of 11 meteorological variables (including temperatures, wind speed and direction, humidity, altimeter) in the US, during the period 2010-2013. We used the 11 variables in our experiments, handling them with channel independence. Sourced by Zhou et al. (2021) from `https://www.ncei.noaa.gov/data/local-climatological-data/`.

**Transport domain**

**Pedestrian Counts\*** contains hourly pedestrian counts captured from 66 sensors in Melbourne city starting from May 2009 and up to 2020. It is part of the Monash Time Series Forecasting Library (Godahewa et al., 2021) and is sourced from the City of Melbourne.

**Traffic** is a collection of 48 months (2015-2016) road occupancy data from the California Department of Transportation (Lai et al., 2018). The road occupancy rates (between 0 and 1) are measured by different sensors on San Francisco Bay area freeways.

**Loop Seattle** contains one-year traffic state data from 323 sensor stations in the Greater Seattle Area, in 2015. Data were collected from inductive loop detectors deployed on four connected freeways (Cui et al., 2019).

**PEMS03** is a highway traffic dataset collected by Song et al. (2020) from the California Department of Transportation Performance Measurement System (PeMS). PEMS03 contains 358 sensors with measurements from January to November 2018. We used four weeks of data for validation, four weeks for testing and the first 35 days for training.

**PEMS BAY** contains six months of measurements of traffic speed from 325 sensors in the Bay Area, California, in 2017 (Li et al., 2018). The train / validation / test split is 0.65 / 0.155 (28 days) / 0.195.

**SHMETRO** contains the passengers inflow and outflow measurements of 288 subway stations in Shanghai for three months in 2016 (Liu et al., 2020). We kept four weeks of data for validation, four weeks for testing and the first 35 days for training.

**M Dense** contains measurements of traffic intensity (number of cars per hour) from 30 sensors located in the city of Madrid, Spain, in 2018-2019. The dataset was sourced by the LibCity library through the open data portal of the Municipality of Madrid (de Medrano & Aznarte, 2020; Jiang et al., 2023).

## B.2   Datasets with covariates

Table 5 shows the key properties of the three datasets used for the evaluation of imputation with additional covariates. Each dataset is split chronologically intro train, validation and test splits with respective fractions

0.7, 0.1 and 0.2. Four-week segments are generated in the same manner as in Section B.1 for the univariate experiments.

Table 5: All datasets with covariates used in our experiments and their key properties.

| Dataset | Release Platform | Freq | Target Series | Covariate | Series Length | Num. Test Segments |
|---------|------------------|------|---------------|-----------|---------------|--------------------|
| PV-France | RTE, Meteo France | 1H | 1 | 1 | 8760 | 38 |
| Wind-France | RTE, Meteo France | 1H | 1 | 1 | 8760 | 38 |
| Load-France | RTE, Enedis | 30min | 1 | 1 | 17520 | 41 |

**PV-France** contains the aggregated photovoltaic (PV) power production (target variable) and the average solar irradiance (covariate) in the southern French region *Occitanie* in 2021. This dataset was obtained by aggregating two sources of data. (i) The target PV power production is provided by France's Transmission System Operator (RTE) and extracted through their data portal `https://www.rte-france.com/en/eco2mix/download-indicators`. (ii) The global solar irradiance is obtained from the French weather institute Meteo France `https://meteo.data.gouv.fr/datasets/donnees-climatologiques-de-base-horaires/`. We aggregated the *in-situ* observations at the department level into a region-level irradiance.

**Wind-France** contains the aggregated wind power production (target variable) and the wind speed (covariate) in the northern French region *Hauts-de-France* in 2021. Similarly to *PV-France*, this dataset is obtained respectively via RTE's data portal for the wind power production and via Meteo France for the wind speed.

**Load-France** contains the total French electricity demand (target variable) and the average temperature (covariate) in 2022. Similarly to *PV-France* and *Wind-France*, the total electricity demand is obtained from RTE's data portal. The national temperature is provided by Enedis, the French distribution grid operator, from `https://data.enedis.fr/explore/dataset/donnees-de-temperature-et-de-pseudo-rayonnement/information/`.

## C  Univariate benchmark: extensive results

### C.1  Full results

Table 6 reports the complete univariate imputation benchmark across all datasets and missingness settings. These detailed results complement the main text by providing per-dataset performance for every method, allowing a finer comparison of their robustness and consistency across different missingness patterns and time series domain. For more details about the datasets please refer to Section B.1. All experiments were carried out on single NVIDIA A100-40G or H100-80G GPUs.

Table 6: Complete univariate benchmark results for all datasets and missingness settings. Best in **bold**, second-best underlined. Settings *Pointwise 1* and *Pointwise 2*: respectively 50% and 70% of observations removed at random. Settings *Blocks 1* and *Blocks 2*: respectively two and four entire days removed at random.

| Dataset | Setting | Foundation models | | Task Specific Models | | | | | | Local Models | | |
|---------|---------|-----------------|------|-------|-------|------|------------------|--------------|-------------|--------|-----------------|------|
| | | TabPFN-TS | MoTM | SAITS | BRITS | CSDI | Time mixerpp | Times net | TSla net | Linear | Seasonal Naive | LOCF |
| **BDG2-Bear** | *Pointwise 1* | **0.171** | 0.202 | 0.241 | 0.309 | 0.234 | 0.827 | 0.829 | 0.833 | 0.229 | 0.532 | 0.391 |
| | *Pointwise 2* | **0.223** | 0.240 | 0.309 | 0.432 | 0.284 | 0.825 | 0.830 | 0.833 | 0.330 | 0.587 | 0.545 |
| | *Blocks 1* | **0.272** | 0.332 | 0.399 | 0.445 | 0.387 | 0.829 | 0.817 | 0.831 | 0.857 | 0.478 | 0.904 |
| | *Blocks 2* | **0.280** | 0.336 | 0.405 | 0.471 | 0.395 | 0.829 | 0.819 | 0.832 | 0.861 | 0.473 | 0.904 |

*Continued on next page*

Table 6 – *Continued from previous page*

| Dataset | Setting | Foundation models | | Task Specific Models | | | | | | Local Models | | |
|---|---|---|---|---|---|---|---|---|---|---|---|---|
| | | TabPFN-TS | MoTM | SAITS | BRITS | CSDI | Time mixerpp | Times net | TS1a net | Linear | Seasonal Naive | LOCF |
| **BDG2-Rat** | *Pointwise 1* | **0.196** | 0.231 | 0.266 | 0.279 | 0.251 | 0.813 | 0.812 | 0.818 | 0.247 | 0.587 | 0.380 |
| | *Pointwise 2* | **0.256** | 0.273 | 0.339 | 0.338 | 0.314 | 0.813 | 0.814 | 0.818 | 0.334 | 0.646 | 0.507 |
| | *Blocks 1* | **0.349** | 0.400 | 0.495 | 0.503 | 0.494 | 0.813 | 0.808 | 0.816 | 0.743 | 0.536 | 0.811 |
| | *Blocks 2* | **0.355** | 0.402 | 0.497 | 0.508 | 0.498 | 0.813 | 0.808 | 0.816 | 0.744 | 0.536 | 0.814 |
| **Borealis** | *Pointwise 1* | 0.417 | 0.519 | **0.403** | 0.407 | 0.687 | 0.598 | 0.535 | 0.596 | 0.442 | 0.647 | 0.543 |
| | *Pointwise 2* | 0.488 | 0.583 | 0.468 | **0.442** | 0.672 | 0.594 | 0.545 | 0.597 | 0.505 | 0.674 | 0.601 |
| | *Blocks 1* | 0.536 | 0.646 | 0.518 | **0.508** | 0.685 | 0.627 | 0.540 | 0.612 | 0.633 | 0.662 | 0.718 |
| | *Blocks 2* | 0.522 | 0.629 | 0.519 | **0.504** | 0.658 | 0.612 | 0.537 | 0.602 | 0.658 | 0.612 | 0.638 |
| **Covid19 Energy** | *Pointwise 1* | **0.075** | 0.099 | 0.399 | 0.307 | 1.113 | 0.405 | 0.414 | 0.852 | 0.163 | 0.438 | 0.387 |
| | *Pointwise 2* | 0.132 | **0.127** | 0.417 | 0.436 | 1.118 | 0.486 | 0.417 | 0.854 | 0.292 | 0.483 | 0.570 |
| | *Blocks 1* | **0.202** | 0.222 | 0.432 | 0.629 | 1.117 | 0.604 | 0.440 | 0.858 | 0.949 | 0.399 | 0.975 |
| | *Blocks 2* | **0.201** | 0.232 | 0.436 | 0.638 | 1.114 | 0.599 | 0.461 | 0.842 | 0.928 | 0.392 | 0.939 |
| **GFC12 Load** | *Pointwise 1* | **0.143** | 0.189 | 0.188 | 0.263 | 0.174 | 0.789 | 0.776 | 0.803 | 0.232 | 0.603 | 0.448 |
| | *Pointwise 2* | 0.237 | **0.231** | 0.280 | 0.457 | 0.248 | 0.789 | 0.787 | 0.804 | 0.370 | 0.670 | 0.600 |
| | *Blocks 1* | **0.353** | 0.382 | 0.417 | 0.524 | 0.428 | 0.793 | 0.788 | 0.797 | 0.810 | 0.546 | 0.868 |
| | *Blocks 2* | **0.363** | 0.387 | 0.425 | 0.541 | 0.436 | 0.799 | 0.796 | 0.803 | 0.814 | 0.546 | 0.871 |
| **Hog** | *Pointwise 1* | **0.196** | 0.240 | 0.245 | 0.244 | 0.279 | 0.585 | 0.785 | 0.802 | 0.216 | 0.701 | 0.322 |
| | *Pointwise 2* | **0.260** | 0.286 | 0.320 | 0.326 | 0.350 | 0.593 | 0.795 | 0.801 | 0.280 | 0.759 | 0.418 |
| | *Blocks 1* | **0.396** | 0.458 | 0.518 | 0.562 | 0.569 | 0.651 | 0.766 | 0.799 | 0.555 | 0.640 | 0.668 |
| | *Blocks 2* | **0.406** | 0.464 | 0.528 | 0.568 | 0.580 | 0.655 | 0.766 | 0.795 | 0.564 | 0.649 | 0.673 |
| **Ideal** | *Pointwise 1* | 0.501 | 0.570 | **0.443** | 0.464 | 0.603 | 0.531 | 0.702 | 0.691 | 0.526 | 0.678 | 0.620 |
| | *Pointwise 2* | 0.571 | 0.644 | **0.473** | 0.504 | 0.638 | 0.572 | 0.706 | 0.692 | 0.592 | 0.701 | 0.683 |
| | *Blocks 1* | 0.558 | 0.667 | **0.498** | 0.543 | 0.655 | 0.531 | 0.706 | 0.696 | 0.730 | 0.650 | 0.724 |
| | *Blocks 2* | 0.564 | 0.658 | **0.485** | 0.522 | 0.648 | 0.497 | 0.688 | 0.676 | 0.699 | 0.657 | 0.744 |
| **PDB** | *Pointwise 1* | **0.062** | 0.094 | 0.337 | 0.408 | 1.122 | 0.449 | 0.395 | 0.864 | 0.191 | 0.375 | 0.435 |
| | *Pointwise 2* | **0.119** | 0.121 | 0.373 | 0.556 | 1.128 | 0.550 | 0.599 | 0.858 | 0.338 | 0.413 | 0.631 |
| | *Blocks 1* | **0.171** | 0.197 | 0.353 | 0.637 | 1.108 | 0.726 | 0.265 | 0.832 | 0.982 | 0.331 | 1.004 |
| | *Blocks 2* | **0.177** | 0.201 | 0.356 | 0.653 | 1.113 | 0.740 | 0.291 | 0.847 | 1.006 | 0.336 | 1.010 |
| **KDD Cup2022** | *Pointwise 1* | 0.099 | 0.237 | 0.115 | 0.764 | 0.171 | 0.764 | 0.758 | 0.775 | **0.092** | 0.955 | 0.139 |
| | *Pointwise 2* | 0.120 | 0.255 | 0.161 | 0.768 | 0.217 | 0.767 | 0.766 | 0.776 | **0.115** | 0.971 | 0.176 |
| | *Blocks 1* | 0.571 | 0.636 | 0.628 | 0.763 | 0.914 | 0.763 | 0.764 | 0.769 | **0.467** | 0.939 | 0.654 |
| | *Blocks 2* | 0.566 | 0.634 | 0.624 | 0.758 | 0.912 | 0.758 | 0.761 | 0.765 | **0.464** | 0.927 | 0.660 |
| **ERA5 geo.** | *Pointwise 1* | **0.091** | 0.168 | 0.149 | 0.161 | 0.163 | 0.251 | 0.813 | 0.815 | 0.104 | 0.728 | 0.224 |
| | *Pointwise 2* | **0.146** | 0.208 | 0.219 | 0.275 | 0.253 | 0.427 | 0.814 | 0.815 | 0.162 | 0.791 | 0.322 |
| | *Blocks 1* | **0.333** | 0.452 | 0.525 | 0.905 | 0.680 | 0.709 | 0.806 | 0.808 | 0.473 | 0.681 | 0.627 |
| | *Blocks 2* | **0.336** | 0.449 | 0.524 | 0.908 | 0.680 | 0.710 | 0.807 | 0.809 | 0.475 | 0.681 | 0.629 |
| **ERA5 humidity** | *Pointwise 1* | **0.129** | 0.212 | 0.169 | 0.177 | 0.216 | 0.413 | 0.788 | 1.670 | 0.131 | 0.664 | 0.240 |
| | *Pointwise 2* | 0.200 | 0.254 | 0.233 | 0.270 | 0.315 | 0.544 | 0.792 | 1.320 | 0.191 | 0.748 | 0.314 |
| | *Blocks 1* | **0.336** | 0.434 | 0.415 | 0.501 | 0.608 | 0.558 | 0.787 | 2.059 | 0.372 | 0.603 | 0.500 |
| | *Blocks 2* | **0.337** | 0.431 | 0.417 | 0.503 | 0.608 | 0.559 | 0.784 | 1.989 | 0.373 | 0.606 | 0.506 |
| **ERA5 temp.** | *Pointwise 1* | **0.094** | 0.168 | 0.158 | 0.157 | 0.177 | 0.481 | 0.817 | 1.084 | 0.112 | 0.700 | 0.237 |
| | *Pointwise 2* | **0.150** | 0.208 | 0.231 | 0.250 | 0.261 | 0.603 | 0.815 | 0.947 | 0.174 | 0.765 | 0.340 |
| | *Blocks 1* | **0.327** | 0.438 | 0.520 | 0.534 | 0.677 | 0.725 | 0.810 | 1.292 | 0.504 | 0.651 | 0.647 |
| | *Blocks 2* | **0.331** | 0.436 | 0.521 | 0.539 | 0.681 | 0.725 | 0.808 | 1.258 | 0.503 | 0.653 | 0.647 |
| **ERA5 wind** | *Pointwise 1* | **0.124** | 0.225 | 0.176 | 0.182 | 0.195 | 0.797 | 0.790 | 3.795 | 0.127 | 0.945 | 0.252 |
| | *Pointwise 2* | 0.201 | 0.281 | 0.248 | 0.298 | 0.285 | 0.793 | 0.795 | 2.637 | 0.187 | 0.994 | 0.346 |
| | *Blocks 1* | **0.461** | 0.604 | 0.552 | 0.706 | 0.722 | 0.802 | 0.800 | 5.137 | 0.465 | 0.902 | 0.685 |
| | *Blocks 2* | **0.464** | 0.604 | 0.553 | 0.713 | 0.720 | 0.799 | 0.797 | 4.954 | 0.467 | 0.910 | 0.687 |
| **Oikolab Weather** | *Pointwise 1* | **0.150** | 0.227 | 0.207 | 0.207 | 0.226 | 0.802 | 0.824 | 0.827 | 0.164 | 0.811 | 0.313 |
| | *Pointwise 2* | **0.228** | 0.278 | 0.294 | 0.327 | 0.321 | 0.805 | 0.826 | 0.828 | 0.248 | 0.857 | 0.436 |
| | *Blocks 1* | **0.449** | 0.529 | 0.581 | 0.669 | 0.739 | 0.820 | 0.815 | 0.823 | 0.638 | 0.766 | 0.786 |
| | *Blocks 2* | **0.454** | 0.534 | 0.584 | 0.674 | 0.735 | 0.821 | 0.818 | 0.826 | 0.636 | 0.770 | 0.787 |
| **Pedestrian Counts** | *Pointwise 1* | **0.150** | 0.196 | 0.240 | 0.268 | 0.176 | 0.786 | 0.804 | 0.812 | 0.329 | 0.347 | 0.526 |
| | *Pointwise 2* | **0.200** | 0.239 | 0.289 | 0.387 | 0.213 | 0.793 | 0.806 | 0.812 | 0.447 | 0.383 | 0.684 |
| | *Blocks 1* | **0.172** | 0.254 | 0.243 | 0.619 | 0.210 | 0.797 | 0.795 | 0.810 | 1.001 | 0.307 | 1.002 |
| | *Blocks 2* | **0.178** | 0.260 | 0.252 | 0.632 | 0.214 | 0.797 | 0.797 | 0.811 | 1.000 | 0.310 | 0.997 |
| **Traffic** | *Pointwise 1* | **0.172** | 0.237 | 0.174 | 0.208 | 0.208 | 0.735 | 0.735 | 0.744 | 0.288 | 0.380 | 0.498 |
| | *Pointwise 2* | **0.216** | 0.285 | 0.228 | 0.242 | 0.242 | 0.738 | 0.738 | 0.744 | 0.421 | 0.416 | 0.670 |
| | *Blocks 1* | **0.204** | 0.301 | 0.235 | 0.252 | 0.252 | 0.731 | 0.731 | 0.744 | 0.985 | 0.340 | 0.985 |
| | *Blocks 2* | **0.210** | 0.307 | 0.242 | 0.256 | 0.256 | 0.732 | 0.732 | 0.744 | 0.986 | 0.341 | 0.985 |
| **SHMETRO** | *Pointwise 1* | **0.123** | 0.297 | 0.530 | 0.299 | 1.413 | 0.642 | 0.314 | 0.675 | 0.168 | 0.254 | 0.275 |
| | *Pointwise 2* | **0.143** | 0.325 | 0.537 | 0.376 | 1.493 | 0.641 | 0.450 | 0.673 | 0.219 | 0.280 | 0.363 |
| | *Blocks 1* | **0.190** | 0.407 | 0.538 | 0.632 | 1.283 | 0.642 | 0.277 | 0.671 | 0.963 | 0.217 | 0.970 |
| | *Blocks 2* | **0.197** | 0.409 | 0.546 | 0.632 | 1.441 | 0.642 | 0.289 | 0.670 | 0.917 | 0.220 | 0.910 |

*Continued on next page*

Table 6 – *Continued from previous page*

| Dataset | Setting | Foundation models | | Task Specific Models | | | | | | Local Models | | |
|---|---|---|---|---|---|---|---|---|---|---|---|---|
| | | TabPFN-TS | MoTM | SAITS | BRITS | CSDI | Time mixerpp | Times net | TS1a net | Linear | Seasonal Naive | LOCF |
| **PEMS03** | *Pointwise 1* | **0.133** | 0.189 | 0.353 | 0.360 | 1.300 | 0.862 | 0.439 | 0.883 | 0.148 | 0.349 | 0.183 |
| | *Pointwise 2* | **0.148** | 0.197 | 0.351 | 0.496 | 1.440 | 0.862 | 0.605 | 0.881 | 0.156 | 0.377 | 0.206 |
| | *Blocks 1* | **0.233** | 0.333 | 0.376 | 0.862 | 1.150 | 0.867 | 0.323 | 0.879 | 1.098 | 0.330 | 1.109 |
| | *Blocks 2* | **0.229** | 0.332 | 0.367 | 0.855 | 1.176 | 0.861 | 0.312 | 0.872 | 1.083 | 0.315 | 1.097 |
| **PEMS BAY** | *Pointwise 1* | 0.127 | 0.306 | 0.188 | 0.195 | 0.863 | 0.558 | 0.595 | 0.642 | **0.121** | 0.458 | 0.170 |
| | *Pointwise 2* | 0.146 | 0.322 | 0.197 | 0.203 | 0.851 | 0.558 | 0.615 | 0.642 | **0.145** | 0.490 | 0.207 |
| | *Blocks 1* | **0.324** | 0.513 | 0.423 | 0.435 | 0.865 | 0.555 | 0.638 | 0.641 | 0.775 | 0.422 | 0.800 |
| | *Blocks 2* | **0.329** | 0.521 | 0.423 | 0.436 | 0.875 | 0.560 | 0.639 | 0.643 | 0.776 | 0.427 | 0.795 |
| **ETT1-15T** | *Pointwise 1* | **0.180** | 0.288 | 0.227 | 0.260 | 0.267 | 0.535 | 0.500 | 0.793 | 0.183 | 0.610 | 0.256 |
| | *Pointwise 2* | **0.208** | 0.311 | 0.262 | 0.357 | 0.322 | 0.585 | 0.619 | 0.793 | 0.213 | 0.648 | 0.316 |
| | *Blocks 1* | **0.445** | 0.504 | 0.662 | 0.760 | 0.688 | 0.783 | 0.626 | 0.791 | 0.821 | 0.584 | 0.890 |
| | *Blocks 2* | **0.452** | 0.513 | 0.660 | 0.759 | 0.698 | 0.780 | 0.648 | 0.790 | 0.814 | 0.581 | 0.883 |
| **ETT1-1H** | *Pointwise 1* | **0.230** | 0.279 | 0.314 | 0.309 | 0.380 | 0.787 | 0.596 | 0.797 | 0.279 | 0.588 | 0.446 |
| | *Pointwise 2* | **0.306** | 0.326 | 0.404 | 0.399 | 0.482 | 0.784 | 0.692 | 0.797 | 0.381 | 0.630 | 0.582 |
| | *Blocks 1* | **0.412** | 0.465 | 0.577 | 0.603 | 0.683 | 0.793 | 0.610 | 0.796 | 0.820 | 0.562 | 0.882 |
| | *Blocks 2* | **0.419** | 0.473 | 0.583 | 0.617 | 0.690 | 0.793 | 0.617 | 0.796 | 0.828 | 0.557 | 0.891 |
| **ETT2-1H** | *Pointwise 1* | **0.281** | 0.321 | 0.333 | 0.341 | 0.450 | 0.778 | 0.758 | 0.801 | 0.293 | 0.698 | 0.426 |
| | *Pointwise 2* | **0.350** | 0.371 | 0.395 | 0.420 | 0.514 | 0.780 | 0.761 | 0.801 | 0.365 | 0.750 | 0.525 |
| | *Blocks 1* | **0.480** | 0.526 | 0.568 | 0.635 | 0.697 | 0.790 | 0.781 | 0.798 | 0.686 | 0.674 | 0.784 |
| | *Blocks 2* | **0.490** | 0.527 | 0.575 | 0.642 | 0.692 | 0.788 | 0.783 | 0.797 | 0.689 | 0.663 | 0.785 |
| **ETT2-15T** | *Pointwise 1* | **0.215** | 0.332 | 0.312 | 0.275 | 0.308 | 0.768 | 0.717 | 0.801 | 0.216 | 0.716 | 0.280 |
| | *Pointwise 2* | **0.247** | 0.356 | 0.339 | 0.351 | 0.368 | 0.768 | 0.781 | 0.801 | 0.249 | 0.764 | 0.334 |
| | *Blocks 1* | **0.543** | 0.554 | 0.592 | 0.756 | 0.713 | 0.792 | 0.777 | 0.799 | 0.675 | 0.688 | 0.765 |
| | *Blocks 2* | **0.547** | 0.553 | 0.594 | 0.759 | 0.718 | 0.797 | 0.783 | 0.805 | 0.697 | 0.691 | 0.785 |
| **Solar-1H** | *Pointwise 1* | 0.132 | **0.131** | 0.141 | 0.297 | 0.140 | 0.304 | 0.632 | 0.828 | 0.216 | 0.260 | 0.407 |
| | *Pointwise 2* | 0.176 | **0.168** | 0.213 | 0.415 | 0.187 | 0.310 | 0.685 | 0.828 | 0.365 | 0.277 | 0.589 |
| | *Blocks 1* | **0.219** | 0.228 | 0.237 | 0.347 | 0.264 | 0.299 | 0.534 | 0.829 | 0.930 | 0.239 | 0.921 |
| | *Blocks 2* | **0.219** | 0.229 | 0.236 | 0.357 | 0.265 | 0.298 | 0.550 | 0.830 | 0.932 | 0.239 | 0.924 |
| **Jena Weather 10T** | *Pointwise 1* | 0.086 | 0.190 | 0.177 | 0.132 | 0.174 | 0.724 | 0.339 | 0.748 | **0.080** | 0.635 | 0.123 |
| | *Pointwise 2* | 0.101 | 0.207 | 0.196 | 0.188 | 0.201 | 0.725 | 0.490 | 0.748 | **0.096** | 0.688 | 0.157 |
| | *Blocks 1* | **0.378** | 0.453 | 0.557 | 0.681 | 0.685 | 0.739 | 0.699 | 0.746 | 0.527 | 0.591 | 0.640 |
| | *Blocks 2* | **0.384** | 0.459 | 0.561 | 0.681 | 0.712 | 0.736 | 0.700 | 0.743 | 0.544 | 0.598 | 0.644 |
| **Jena Weather 1H** | *Pointwise 1* | **0.156** | 0.224 | 0.346 | 0.223 | 0.323 | 0.738 | 0.675 | 0.754 | 0.170 | 0.638 | 0.293 |
| | *Pointwise 2* | **0.214** | 0.272 | 0.394 | 0.316 | 0.418 | 0.741 | 0.725 | 0.754 | 0.243 | 0.700 | 0.394 |
| | *Blocks 1* | **0.366** | 0.463 | 0.532 | 0.583 | 0.674 | 0.736 | 0.703 | 0.744 | 0.552 | 0.608 | 0.662 |
| | *Blocks 2* | **0.370** | 0.464 | 0.532 | 0.591 | 0.679 | 0.744 | 0.711 | 0.750 | 0.555 | 0.606 | 0.653 |
| **Loop Seattle 5T** | *Pointwise 1* | **0.233** | 0.371 | 0.326 | 0.345 | 0.309 | 0.661 | 0.708 | 0.721 | 0.246 | 0.644 | 0.300 |
| | *Pointwise 2* | **0.261** | 0.387 | 0.345 | 0.368 | 0.334 | 0.661 | 0.708 | 0.721 | 0.266 | 0.672 | 0.331 |
| | *Blocks 1* | **0.469** | 0.590 | 0.596 | 0.665 | 0.573 | 0.662 | 0.709 | 0.722 | 0.842 | 0.611 | 0.895 |
| | *Blocks 2* | **0.474** | 0.597 | 0.600 | 0.675 | 0.574 | 0.662 | 0.710 | 0.722 | 0.842 | 0.612 | 0.898 |
| **Loop Seattle 1H** | *Pointwise 1* | **0.279** | 0.337 | 0.321 | 0.402 | 0.325 | 0.505 | 0.715 | 0.721 | 0.379 | 0.576 | 0.530 |
| | *Pointwise 2* | **0.350** | 0.394 | 0.376 | 0.502 | 0.373 | 0.524 | 0.713 | 0.721 | 0.486 | 0.615 | 0.653 |
| | *Blocks 1* | **0.345** | 0.417 | 0.367 | 0.511 | 0.394 | 0.505 | 0.715 | 0.720 | 0.845 | 0.532 | 0.884 |
| | *Blocks 2* | **0.354** | 0.425 | 0.376 | 0.523 | 0.400 | 0.511 | 0.717 | 0.721 | 0.851 | 0.533 | 0.890 |
| **M Dense** | *Pointwise 1* | **0.209** | 0.233 | 0.274 | 0.422 | 0.276 | 0.851 | 0.456 | 0.855 | 0.341 | 0.447 | 0.552 |
| | *Pointwise 2* | **0.252** | 0.276 | 0.315 | 0.604 | 0.316 | 0.854 | 0.634 | 0.855 | 0.473 | 0.484 | 0.722 |
| | *Blocks 1* | **0.225** | 0.284 | 0.288 | 0.415 | 0.302 | 0.849 | 0.433 | 0.856 | 1.039 | 0.405 | 1.060 |
| | *Blocks 2* | **0.228** | 0.288 | 0.291 | 0.433 | 0.305 | 0.847 | 0.455 | 0.854 | 1.041 | 0.407 | 1.061 |
| **Enedis LDM Small** | *Pointwise 1* | 0.340 | 0.480 | **0.299** | 0.317 | 0.356 | 0.499 | 0.584 | 0.603 | 0.340 | 0.373 | 0.439 |
| | *Pointwise 2* | 0.440 | 0.534 | **0.338** | 0.358 | 0.415 | 0.499 | 0.589 | 0.594 | 0.419 | 0.396 | 0.524 |
| | *Blocks 1* | 0.358 | 0.748 | **0.337** | 0.358 | 0.451 | 0.496 | 0.581 | 0.637 | 0.647 | 0.351 | 0.719 |
| | *Blocks 2* | 0.366 | 0.515 | **0.341** | 0.362 | 0.456 | 0.498 | 0.581 | 0.633 | 0.707 | 0.345 | 0.722 |
| **London Small** | *Pointwise 1* | 0.439 | 0.573 | **0.432** | 0.434 | 0.837 | 0.596 | 0.650 | 0.965 | 0.480 | 0.666 | 0.573 |
| | *Pointwise 2* | 0.490 | 0.632 | **0.465** | 0.483 | 0.843 | 0.596 | 0.651 | 0.760 | 0.534 | 0.679 | 0.638 |
| | *Blocks 1* | **0.521** | 0.622 | 0.537 | 0.571 | 0.822 | 0.595 | 0.650 | 1.631 | 0.794 | 0.656 | 0.836 |
| | *Blocks 2* | 0.528 | 0.628 | **0.512** | 0.576 | 0.825 | 0.595 | 0.650 | 1.570 | 0.798 | 0.656 | 0.836 |
| **Spanish Energy** | *Pointwise 1* | **0.131** | 0.200 | 0.200 | 0.235 | 1.716 | 0.782 | 0.812 | 0.802 | 0.164 | 0.680 | 0.298 |
| | *Pointwise 2* | **0.207** | 0.246 | 0.290 | 0.346 | 1.905 | 0.783 | 0.810 | 0.802 | 0.253 | 0.735 | 0.413 |
| | *Blocks 1* | **0.400** | 0.472 | 0.559 | 0.633 | 1.367 | 0.787 | 0.788 | 0.801 | 0.614 | 0.642 | 0.719 |
| | *Blocks 2* | **0.399** | 0.469 | 0.559 | 0.637 | 1.384 | 0.784 | 0.788 | 0.800 | 0.609 | 0.635 | 0.715 |
| **Weather** | *Pointwise 1* | **0.257** | 0.298 | 0.290 | 0.360 | 3.369 | 0.801 | 0.804 | 0.804 | 0.259 | 0.739 | 0.373 |
| | *Pointwise 2* | **0.311** | 0.342 | 0.364 | 0.458 | 3.928 | 0.792 | 0.803 | 0.804 | 0.322 | 0.803 | 0.472 |
| | *Blocks 1* | **0.456** | 0.494 | 0.592 | 0.689 | 1.680 | 0.803 | 0.797 | 0.800 | 0.620 | 0.686 | 0.736 |
| | *Blocks 2* | **0.455** | 0.488 | 0.590 | 0.691 | 1.909 | 0.806 | 0.799 | 0.801 | 0.618 | 0.682 | 0.735 |

*Continued on next page*

Table 6 – *Continued from previous page*

| Dataset | Setting | Foundation models | | Task Specific Models | | | | | | Local Models | | |
|---|---|---|---|---|---|---|---|---|---|---|---|---|
| | | TabPFN -TS | MoTM | SAITS | BRITS | CSDI | Time mixerpp | Times net | TS1a net | Linear | Seasonal Naive | LOCF |
| **Mean (Std)** | | **0.293** **(0.138)** | 0.371 (0.153) | 0.386 (0.141) | 0.470 (0.182) | 0.664 (0.553) | 0.677 (0.144) | 0.677 (0.150) | 0.930 (0.644) | 0.506 (0.287) | 0.581 (0.180) | 0.611 (0.252) |

## C.2 Performance breakdown by missingness pattern

Figure 7 provides a detailed breakdown of the univariate benchmark results presented in the main paper, focusing on the four distinct missingness settings introduced in Section 3.1. Specifically, we report aggregated z-normalized MAE scores for (a) 50% and (b) 70% pointwise missingness, as well as for (c) two-day and (d) four-day block missingness. These complementary analyses aim to highlight the robustness and consistency of model performance across different temporal corruption patterns.

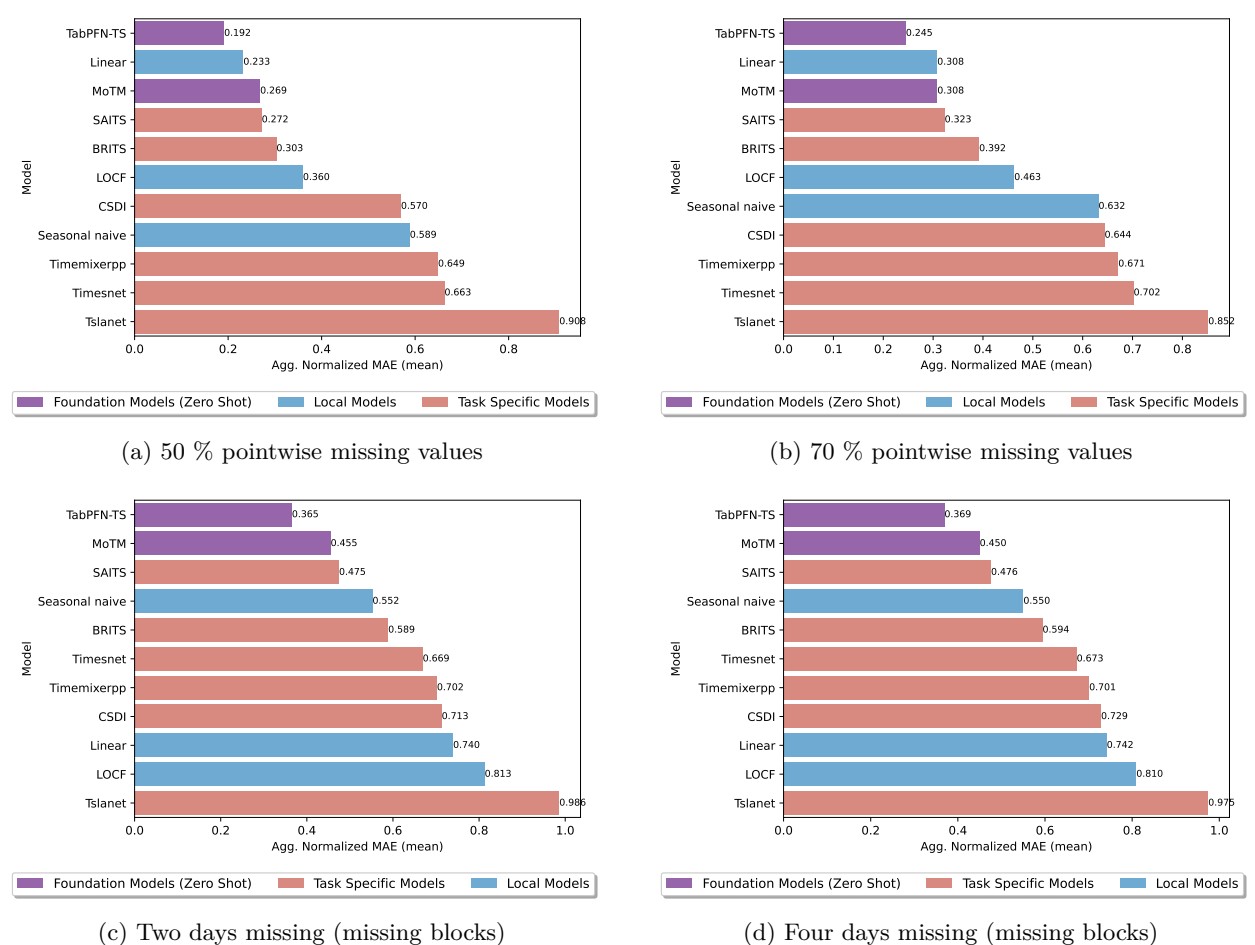

(a) 50 % pointwise missing values

(b) 70 % pointwise missing values

(c) Two days missing (missing blocks)

(d) Four days missing (missing blocks)

Figure 7: Normalized MAEs results for each distinct setting of missingness patterns.

**Results.** The per-setting breakdown shows a consistent advantage of `TabPFN-TS` across all four settings. However, the relative ranking of other methods varies with the missingness pattern: in the first pointwise setting (50%) `Linear` is actually stronger than `MoTM` and remains roughly tied with `MoTM` at 70% pointwise, indicating that simple local interpolation can excel when gaps are sparse. By contrast, in block-missing scenarios (two and four days) `MoTM` substantially outperforms `Linear`, showing superior robustness to structured, long gaps. `SAITS` is the best supervised baseline but is generally behind the foundation models;

its relative position is closer to `MoTM` in some settings (e.g. pointwise). Overall, these results emphasize (i) the stability of `TabPFN-TS` across patterns, (ii) that local methods can be competitive for sparse pointwise missingness, and (iii) that foundation/zero-shot models better handle large or by-block gaps.

### C.3 Quantile predictions

In addition to pointwise imputations, we evaluate the probabilistic capabilities of the time-indexed foundation models by assessing the quality of their predicted quantiles. This analysis complements the deterministic metrics by measuring how well models can represent uncertainty around missing points. Accurate quantile estimation is particularly important for time-series imputation under distributional shifts or irregular sampling, as it reflects the model's ability to produce calibrated and reliable uncertainty quantification.

#### C.3.1 Implementation details

**Weighted Quantile Loss (WQL) definition.** We evaluate probabilistic imputations using the Weighted Quantile Loss (WQL), introduced by Koenker & Hallock (2001) and adopted in probabilistic forecasting benchmarks such as Gneiting et al. (2007); Gasthaus et al. (2019). Given a predicted $\alpha$-quantile $q$ of an observation $x$, the quantile loss is defined as:

$$
\mathrm{QL}_\alpha(q, x) = \begin{cases} \alpha(x - q), & \text{if } x > q, \\ (1 - \alpha)(q - x), & \text{otherwise.} \end{cases} \tag{1}
$$

To aggregate this quantity across all series and time steps, we compute a weighted average normalized by the absolute scale of targets:

$$
\mathrm{WQL}_\alpha = \frac{2 \sum_{i,t} \mathrm{QL}_\alpha(q_{i,t}^{(\alpha)}, x_{i,t})}{\sum_{i,t} |x_{i,t}|}. \tag{2}
$$

We then average over a finite set of quantile levels $\{\alpha_1, \ldots, \alpha_K\}$:

$$
\mathrm{WQL} = \frac{1}{K} \sum_{j=1}^{K} \mathrm{WQL}_{\alpha_j}. \tag{3}
$$

Quantiles are evaluated at $\alpha \in \{0.1, 0.2, \ldots, 0.9\}$ ($K = 9$). Being a weighted average of quantile losses across levels, WQL approximates the Continuous Ranked Probability Score (CRPS), a standard metric for probabilistic accuracy.

**CSDI and SAITS for quantile evaluation.** For `CSDI` (a stochastic diffusion imputer), we estimate quantiles *post hoc* by Monte Carlo sampling at test time. Conditioned on the observed context and the evaluation mask, we draw $S$ imputations $\{\tilde{x}_{i,t}^{(s)}\}_{s=1}^{S}$ from the model's posterior and compute empirical quantiles per index $(i,t)$: $\hat{q}_{i,t}^{(\alpha)} = \mathrm{Quantile}_\alpha\left(\{\tilde{x}_{i,t}^{(s)}\}_{s=1}^{S}\right)$. We use $S = 50$ across all datasets. For `SAITS`, we train *one independent model per quantile level* $\alpha \in \{0.1, \ldots, 0.9\}$. Each model uses the standard pinball (quantile) loss computed only on the evaluation mask, with the same masking patterns as in the pointwise setting. At inference, we obtain the full set of quantiles by stacking the predictions from the $K$ independently trained models. Both models are evaluated on identical imputation splits and masking ratios; WQL is computed only on masked targets and normalized by the absolute scale as described in Eq. (2).

#### C.3.2 Quantitative results

**Setting.** We evaluate the models for uncertainty quantification on 11 univariate datasets, namely: *BDG2-Bear & Rat, Covid19 Energy, GFC12 Load, Hog, Jena Weather 10T, Jena Weather 1H, Oikolab Weather, PDB, Pedestrian Counts* and *Weather*, described in Section B. All models are trained with the same masking ratios and evaluated on the same imputation splits to ensure comparability.

Table 7: Complete WQL results on 11 datasets. For each setting, the best score (lowest) is in **bold** and the second best is underlined.

| Dataset | Setting | TabPFN-TS | MoTM | CSDI | SAITS-adapted |
|---|---|---|---|---|---|
| **BDG2-Bear** | *Pointwise 1* | **0.165** | 0.208 | 0.183 | 0.294 |
| | *Pointwise 2* | **0.208** | 0.256 | 0.223 | 0.373 |
| | *Blocks 1* | **0.253** | 0.354 | 0.336 | 0.493 |
| | *Blocks 2* | **0.257** | 0.355 | 0.338 | 0.493 |
| **BDG2-Rat** | *Pointwise 1* | **0.191** | 0.239 | 0.194 | 0.321 |
| | *Pointwise 2* | **0.203** | 0.256 | 0.242 | 0.412 |
| | *Blocks 1* | **0.338** | 0.436 | 0.387 | 0.617 |
| | *Blocks 2* | **0.334** | 0.431 | 0.382 | 0.612 |
| **Covid19 Energy** | *Pointwise 1* | **0.071** | 0.097 | 0.758 | 0.353 |
| | *Pointwise 2* | **0.124** | 0.130 | 0.757 | 0.418 |
| | *Blocks 1* | **0.187** | 0.236 | 0.767 | 0.522 |
| | *Blocks 2* | **0.187** | 0.246 | 0.759 | 0.521 |
| **GFC12 Load** | *Pointwise 1* | **0.142** | 0.192 | 0.148 | 0.258 |
| | *Pointwise 2* | 0.212 | 0.249 | **0.233** | 0.371 |
| | *Blocks 1* | **0.349** | 0.427 | 0.383 | 0.528 |
| | *Blocks 2* | **0.349** | 0.423 | 0.381 | 0.523 |
| **Hog** | *Pointwise 1* | **0.196** | 0.249 | 0.230 | 0.322 |
| | *Pointwise 2* | **0.254** | 0.311 | 0.289 | 0.410 |
| | *Blocks 1* | **0.395** | 0.513 | 0.505 | 0.698 |
| | *Blocks 2* | **0.392** | 0.507 | 0.504 | 0.695 |
| **Jena Weather 10T** | *Pointwise 1* | **0.113** | 0.225 | 0.141 | 0.418 |
| | *Pointwise 2* | **0.134** | 0.235 | 0.164 | 0.445 |
| | *Blocks 1* | **0.397** | 0.508 | 0.706 | 0.791 |
| | *Blocks 2* | **0.394** | 0.500 | 0.712 | 0.781 |
| **Jena Weather 1H** | *Pointwise 1* | **0.181** | 0.254 | 0.246 | 0.354 |
| | *Pointwise 2* | **0.234** | 0.313 | 0.317 | 0.437 |
| | *Blocks 1* | **0.384** | 0.548 | 0.619 | 0.697 |
| | *Blocks 2* | **0.382** | 0.528 | 0.605 | 0.693 |
| **Oikolab Weather** | *Pointwise 1* | **0.145** | 0.225 | 0.157 | 0.248 |
| | *Pointwise 2* | **0.212** | 0.291 | 0.234 | 0.358 |
| | *Blocks 1* | **0.431** | 0.584 | 0.578 | 0.761 |
| | *Blocks 2* | **0.420** | 0.572 | 0.568 | 0.748 |
| **PDB** | *Pointwise 1* | **0.057** | 0.084 | 0.766 | 0.272 |
| | *Pointwise 2* | **0.111** | 0.118 | 0.764 | 0.347 |
| | *Blocks 1* | **0.170** | 0.209 | 0.774 | 0.391 |
| | *Blocks 2* | **0.171** | 0.205 | 0.773 | 0.406 |
| **Pedestrian Counts** | *Pointwise 1* | **0.147** | 0.201 | 0.171 | 0.228 |
| | *Pointwise 2* | **0.189** | 0.253 | 0.221 | 0.300 |
| | *Blocks 1* | **0.170** | 0.260 | 0.199 | 0.277 |
| | *Blocks 2* | **0.173** | 0.264 | 0.203 | 0.279 |
| **Weather** | *Pointwise 1* | **0.252** | 0.301 | 0.263 | 0.378 |
| | *Pointwise 2* | **0.303** | 0.356 | 0.322 | 0.456 |
| | *Blocks 1* | **0.448** | 0.528 | 0.601 | 0.734 |
| | *Blocks 2* | **0.431** | 0.510 | 0.582 | 0.716 |
| **Mean (Std)** | | **0.250 (0.112)** | 0.334 (0.146) | 0.431 (0.244) | 0.485 (0.178) |

**Results.** Table 7 reports the WQL scores across all eleven datasets and four missingness settings. TabPFN-TS achieves the lowest WQL across most datasets, showing its ability to produce both accurate and well-calibrated uncertainty estimates. MoTM follows with higher WQL but consistent behavior across datasets and settings. CSDI ranks third overall, with performances illustrating its limited generalization abilities: (i) consistently with its training procedure, good scores are obtained in the pointwise settings; (ii) except on small-size training sets, such as *Covid19 Energy* or *PDB*; (iii) CSDI generally suffers from significantly higher losses on the unseen *Block* settings. Finally, the SAITS variant adapted for quantile prediction yields the highest WQL in most settings, highlighting the shortcomings of this straightforward per-quantile adaptation compared with models expressly designed for probabilistic imputation.

### C.3.3 Qualitative results

In Figure 8 and Figure 9, we present qualitative examples of quantile-based imputations for both TabPFN-TS and MoTM on segments where 70% of the values were removed in a pointwise manner.

For each plot, we display the median prediction together with the inter-quantile ranges $[5, 95]$ and $[25, 75]$, which highlight different levels of predictive uncertainty.

**Results.** The visualizations in Figure 8 and Figure 9 confirm that both `TabPFN-TS` and `MoTM` provide effective quantile-based imputations, successfully reconstructing the signal while providing meaningful uncertainty estimates. `TabPFN-TS` particularly excels at generating high-fidelity reconstructions that closely follow the ground truth, capturing fine-grained temporal details and high-frequency oscillations. Its strength lies in its highly adaptive uncertainty quantification; on the volatile *Borealis* dataset, the quantile ranges adeptly widen to reflect increased predictive uncertainty around sharp peaks, demonstrating a sophisticated understanding of local signal dynamics. `MoTM`, for its part, also delivers strong performance by producing robust, albeit smoother, imputations that effectively capture the main trends and cyclical patterns in datasets like *Hog* and *Era5*. Its more regular uncertainty bands provide a consistent and reliable confidence envelope around the reconstructed signal. In summary, while both models prove highly competent for this task, they exhibit different strengths: `TabPFN-TS` favors a detailed, high-fidelity reconstruction, whereas `MoTM` prioritizes capturing the underlying trend with stable uncertainty.

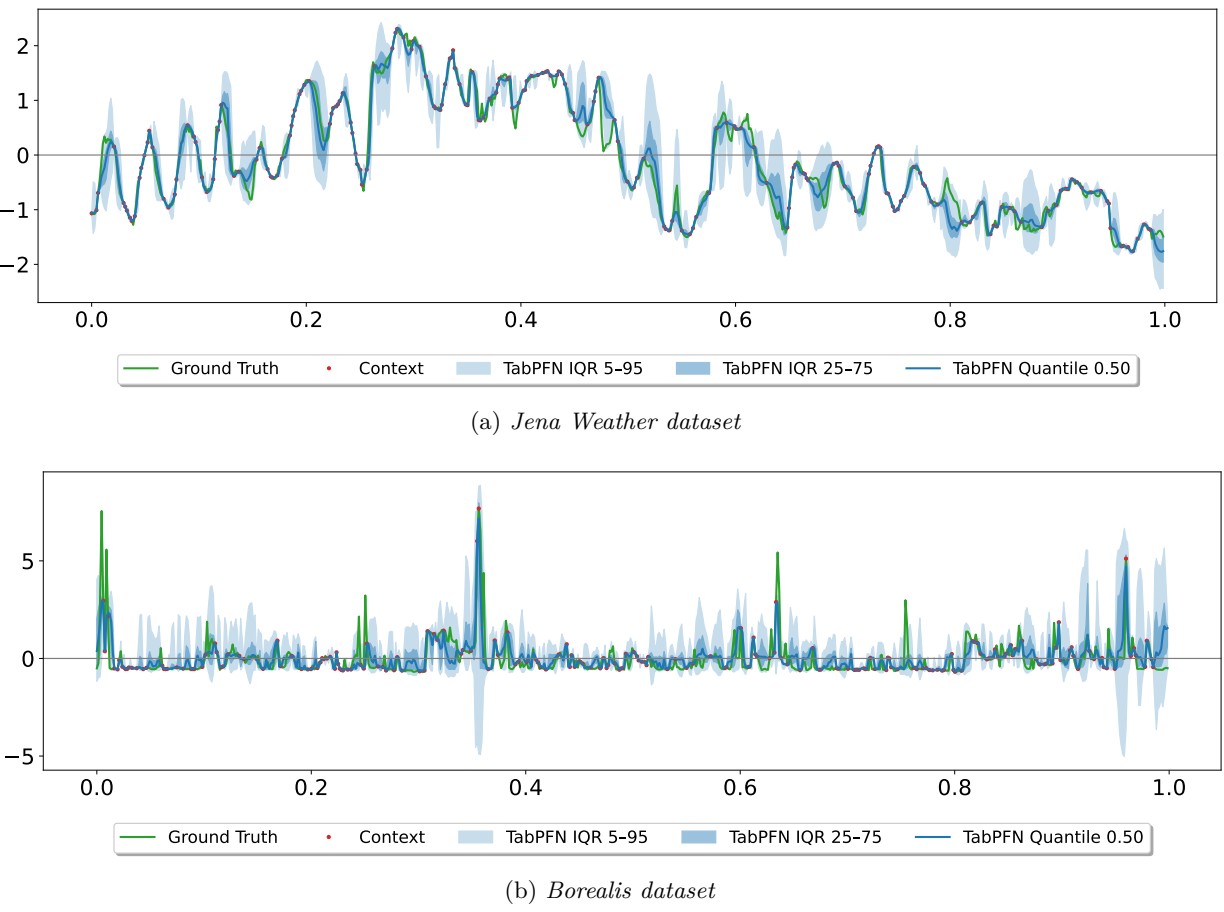

(a) *Jena Weather dataset*

(b) *Borealis dataset*

Figure 8: `TabPFN-TS` qualitative quantile results in the 70% missing values scenario (*Pointwise 2*).

## C.4 Experiments on lower sampling rates datasets

The experiments presented in Section 3.1 focus on datasets with relatively high temporal resolutions (5min, 10min, 15min, 30min, 1h). In this section, we investigate whether time-index foundation models can generalize to significantly lower sampling rates, such as daily or weekly observations. This setup evaluates their robustness to long-term dependencies and coarser temporal granularity, which are common in macroeconomic, energy, or demographic data.

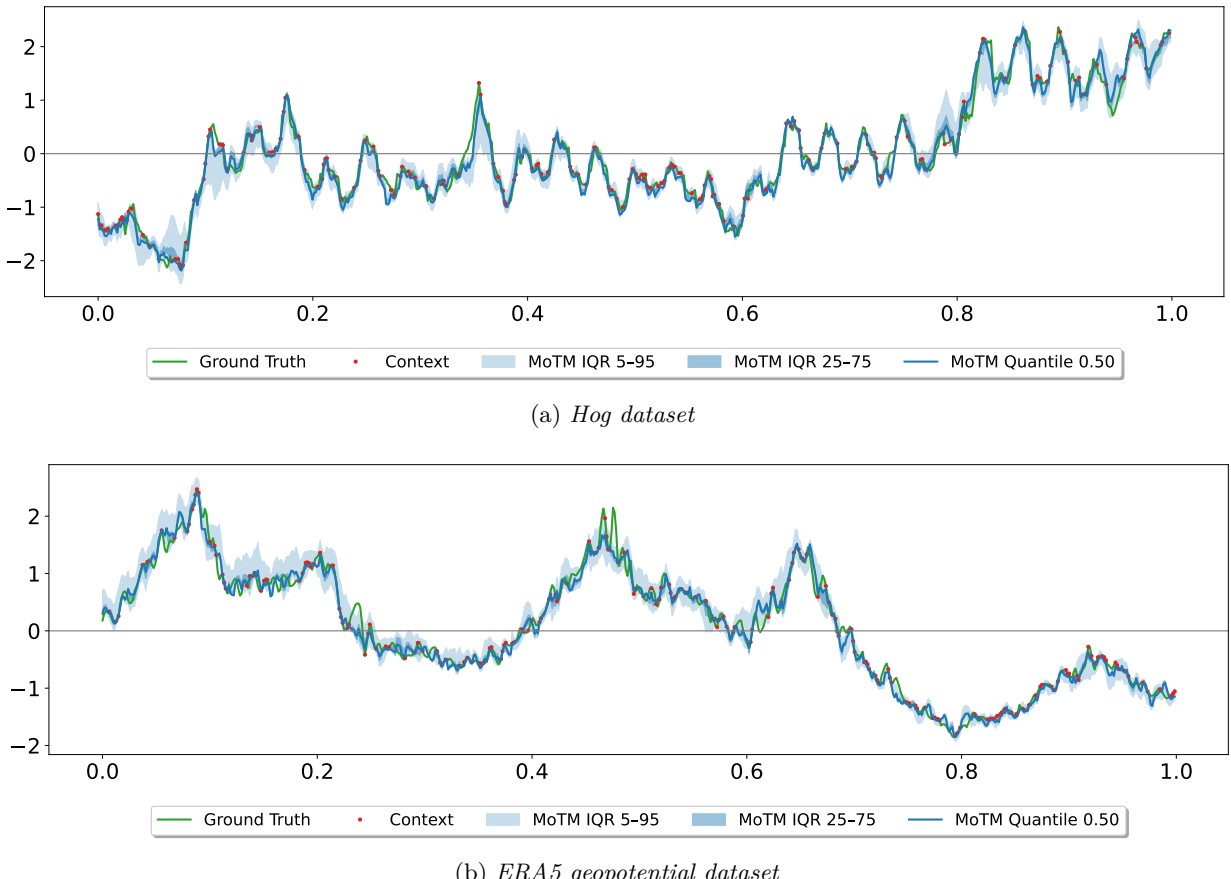

(a) *Hog dataset*

(b) *ERA5 geopotential dataset*

Figure 9: `MoTM` qualitative quantile results in the 70% missing values scenario (*Pointwise 2*).

**Datasets.** We consider four publicly available low-frequency datasets: *Births Daily*, *M4 Daily*, *Births Weekly*, and *M4 Weekly*, with statistics summarized in Table 8. They are part of the Monash Time Series Forecasting archive (Godahewa et al., 2021), and were downloaded from the *GIFT-eval* repository (Aksu et al., 2024). Each dataset exhibits distinct temporal behaviors—seasonality and periodicity are typically weaker at weekly scales, while daily data contain more regular cycles and higher variance. For each dataset, we apply the same four missingness regimes as in the main benchmark (two pointwise and two block-based scenarios), allowing a consistent comparison across frequencies.

Table 8: All datasets at lower sampling rates used in our experiments and their key properties.

| Dataset | Release Platform | Domain | Freq | Num. Series | Series Length | Num. Test Segments |
|---|---|---|---|---|---|---|
| Births Daily | GIFT-eval | Demo. | 1D | 1 | 3652 | 104 |
| Births Weekly | GIFT-eval | Demo. | 1W | 1 | 1043 | 7 |
| M4 Daily | GIFT-eval | Econ./Demo. | 1D | 2112 | 2954 | 80256 |
| M4 Weekly | GIFT-eval | Econ./Demo | 1W | 172 | 947 | 860 |

**Births Daily** contains the daily number of births in the US between 1969 and 1988, as extracted from the R package `mosaicData` (Pruim et al., 2020). **Births Weekly** aggregates these statistics at a weekly frequency.

The M4 Forecasting Competition dataset contains a total of 100k time series, with six different frequencies and from diverse domains such as demography, macroeconomic, etc. (Makridakis et al., 2018; 2020). We used the subsets of, respectively, daily (**M4 Daily**) and weekly (**M4 Weekly**) time series.

**Baselines and settings.** We evaluate a representative subset of imputation methods present in the main benchmark: local heuristics (`Linear`, `Seasonal Naive`), supervised models `SAITS`, `CSDI`, `BRITS`, and the two foundation models `TabPFN-TS` and `MoTM`. As before, `TabPFN-TS` and `MoTM` operate in a fully zero-shot setting, while `SAITS` is retrained for each dataset. All metrics are aggregated using the z-normalized Mean Absolute Error (MAE), and results are reported per dataset and missingness pattern in Table 9.

Table 9: Complete MAE results on datasets with low sampling rates (daily or weekly). Best results are in **bold**, second-best are underlined.

| Dataset | Setting | TabPFN-TS | MoTM | SAITS | BRITS | CSDI | Linear | Seasonnal Naive |
|---|---|---|---|---|---|---|---|---|
| **Births Daily** | *Pointwise 1* | **0.281** | 0.359 | 0.804 | 0.324 | 1.049 | 0.880 | 0.286 |
| | *Pointwise 2* | 0.340 | 0.409 | 0.814 | 0.454 | 1.095 | 0.941 | **0.318** |
| | *Blocks 1* | **0.243** | 0.361 | 0.770 | 0.430 | 1.052 | 0.995 | 0.300 |
| | *Blocks 2* | **0.272** | 0.361 | 0.783 | 0.447 | 1.064 | 0.988 | 0.291 |
| **M4 Daily** | *Pointwise 1* | 0.182 | 0.248 | 0.185 | 0.211 | 0.223 | **0.159** | 0.507 |
| | *Pointwise 2* | 0.246 | 0.304 | 0.234 | 0.288 | 0.273 | **0.195** | 0.606 |
| | *Blocks 1* | 0.442 | 0.531 | 0.430 | 0.576 | 0.638 | **0.381** | 0.547 |
| | *Blocks 2* | 0.454 | 0.534 | 0.441 | 0.591 | 0.614 | **0.385** | 0.559 |
| **Births Weekly** | *Pointwise 1* | **0.293** | 0.337 | 0.896 | 0.748 | 11.999 | 0.332 | 0.492 |
| | *Pointwise 2* | **0.335** | 0.358 | 0.917 | 0.778 | 14.635 | 0.385 | 0.602 |
| | *Blocks 1* | **0.301** | 0.320 | 1.036 | 0.840 | 4.008 | 0.723 | 0.481 |
| | *Blocks 2* | **0.324** | 0.338 | 0.802 | 0.713 | 5.458 | 0.762 | 0.502 |
| **M4 Weekly** | *Pointwise 1* | **0.160** | 0.226 | 0.191 | 0.310 | 0.307 | 0.175 | 0.761 |
| | *Pointwise 2* | **0.188** | 0.254 | 0.226 | 0.415 | 0.344 | 0.198 | 0.836 |
| | *Blocks 1* | **0.244** | 0.326 | 0.430 | 0.635 | 0.435 | 0.318 | 0.643 |
| | *Blocks 2* | **0.256** | 0.345 | 0.444 | 0.643 | 0.459 | 0.340 | 0.650 |
| **Mean (Std)** | | **0.285 (0.089)** | 0.351 (0.097) | 0.525 (0.316) | 0.531 (0.204) | 2.790 (4.323) | 0.512 (0.320) | 0.524 (0.176) |

**Results.** As shown in Table 9, our foundation models demonstrate robust performance on low-frequency data. `TabPFN-TS` achieves the best average score (0.285), followed by `MoTM` (0.351), confirming their ability to generalize to coarser temporal structures without retraining. Other baselines, including specialized models like `SAITS` and `BRITS`, are significantly outperformed and show no clear advantage over simpler methods, while `CSDI` struggles notably. Overall, these results highlight the superior generalization of our zero-shot models in low-frequency regimes compared to both classic baselines and other deep learning architectures.

## D    `TabPFN-TS` **for imputation: additional analysis**

### D.1    `TabPFN-TS` **sensitivity to feature design**

As shown in Figure 1 and Table 2, `TabPFN-TS` delivers remarkably strong zero-shot imputation performance, both in the purely univariate setting and when auxiliary covariates are provided. However, one limitation of the current formulation is that the time-index features fed to the model are manually designed (Cai et al., 2025). This raises an important question: *to what extent does* `TabPFN-TS` *rely on these specific features, and how sensitive is its performance to alternative time encodings?* This section investigates this question by systematically varying the feature set and quantifying its impact on imputation quality.

**Setting.**    We evaluate five alternative feature configurations across 11 representative datasets and the four missingness scenarios used throughout the benchmark. Each configuration corresponds to a different encoding of the time index:

- `TabPFN1`: the original feature set used in the paper, combining the raw grid-normalized time index with daily and weekly sinusoidal encodings:

$$H(t) = \left(t,\ \sin\left(\tfrac{2\pi t}{P_{\text{day}}}\right),\ \cos\left(\tfrac{2\pi t}{P_{\text{day}}}\right),\ \sin\left(\tfrac{2\pi t}{P_{\text{week}}}\right),\ \cos\left(\tfrac{2\pi t}{P_{\text{week}}}\right)\right).$$

- `TabPFN2`: daily-only sinusoidal encodings:

$$H(t) = \left(\sin\left(\tfrac{2\pi t}{P_{\text{day}}}\right),\ \cos\left(\tfrac{2\pi t}{P_{\text{day}}}\right)\right).$$

- `TabPFN3`: raw grid-normalized time index only:

$$H(t) = (t).$$

- `TabPFN4`: purely random periodic features using random periods $P_{\text{rand}_1}$ and $P_{\text{rand}_2}$:

$$H(t) = \left(\sin\left(\tfrac{2\pi t}{P_{\text{rand}_1}}\right),\ \cos\left(\tfrac{2\pi t}{P_{\text{rand}_1}}\right),\ \sin\left(\tfrac{2\pi t}{P_{\text{rand}_2}}\right),\ \cos\left(\tfrac{2\pi t}{P_{\text{rand}_2}}\right)\right).$$

- `TabPFN5`: the original features augmented with the random features used in `TabPFN4`:

$$H(t) = \left(t,\ \sin(\tfrac{2\pi t}{P_{\text{day}}}),\ \cos(\tfrac{2\pi t}{P_{\text{day}}}),\ \sin(\tfrac{2\pi t}{P_{\text{week}}}),\ \cos(\tfrac{2\pi t}{P_{\text{week}}}),\ \sin(\tfrac{2\pi t}{P_{\text{rand}_1}}),\ \cos(\tfrac{2\pi t}{P_{\text{rand}_1}}),\ \sin(\tfrac{2\pi t}{P_{\text{rand}_2}}),\ \cos(\tfrac{2\pi t}{P_{\text{rand}_2}})\right)$$

Each variant probes a distinct aspect of the temporal encoding: (i) removing weekly structure and time index (`TabPFN2`) tests whether multi-frequency seasonality and the relative time index information are essential; (ii) removing the raw time index (`TabPFN2`, `TabPFN4`) evaluates whether relative positional information is necessary; (iii) retaining only the grid index (`TabPFN3`) probes how much the model can extrapolate without any periodic cues; (iv) replacing meaningful periodicities with random ones (`TabPFN4`) tests the robustness of learned priors to feature misspecification; (v) adding spurious features (`TabPFN5`) assesses whether irrelevant periodicities degrade performance by confusing the model's inductive bias. Together, these experiments provide a systematic view of how much `TabPFN-TS` depends on feature engineering versus its own learned prior.

**Setting.**    These variants are evaluated on eleven univariate datasets, namely: *BDG2-Bear* & *Rat, Covid19 Energy, GFC12 Load, Hog, Jena Weather 10T, Jena Weather 1H, Oikolab Weather, PDB, Pedestrian Counts* and *Weather*, with details given in Section B. Table 10 reports the complete MAE results across all eleven datasets and four missingness scenarios.

Table 10: Complete MAE results of five `TabPFN-TS` temporal encoding variants on 11 datasets. For each setting, the best score (lowest) is in **bold** and the second best is underlined.

| Dataset | Setting | TabPFN1 | TabPFN2 | TabPFN3 | TabPFN4 | TabPFN5 |
|---|---|---|---|---|---|---|
| **BDG2-Bear** | *Pointwise 1* | **0.171** | 0.502 | 0.234 | 0.942 | **0.171** |
| | *Pointwise 2* | **0.223** | 0.552 | 0.317 | 0.971 | 0.229 |
| | *Blocks 1* | **0.272** | 0.498 | 0.763 | 0.874 | 0.287 |
| | *Blocks 2* | **0.280** | 0.492 | 0.779 | 0.899 | 0.296 |
| **BDG2-Rat** | *Pointwise 1* | **0.196** | 0.921 | 0.261 | 0.539 | 0.198 |
| | *Pointwise 2* | **0.256** | 0.607 | 0.332 | 0.956 | 0.263 |
| | *Blocks 1* | **0.349** | 0.528 | 0.680 | 0.872 | 0.366 |
| | *Blocks 2* | **0.355** | 0.524 | 0.690 | 0.895 | 0.366 |
| **Covid19 Energy** | *Pointwise 1* | **0.075** | 0.397 | 0.171 | 0.969 | 0.078 |
| | *Pointwise 2* | **0.132** | 0.454 | 0.284 | 1.004 | 0.134 |
| | *Blocks 1* | **0.202** | 0.363 | 0.774 | 0.909 | 0.203 |
| | *Blocks 2* | **0.201** | 0.365 | 0.784 | 0.936 | 0.204 |
| **GFC12 Load** | *Pointwise 1* | **0.143** | 0.537 | 0.280 | 0.935 | 0.144 |
| | *Pointwise 2* | **0.237** | 0.615 | 0.386 | 0.969 | **0.237** |
| | *Blocks 1* | **0.353** | 0.521 | 0.703 | 0.864 | 0.354 |
| | *Blocks 2* | **0.363** | 0.519 | 0.719 | 0.903 | 0.364 |
| **Hog** | *Pointwise 1* | **0.196** | 0.635 | 0.230 | 0.879 | **0.196** |
| | *Pointwise 2* | **0.260** | 0.716 | 0.287 | 0.924 | 0.261 |
| | *Blocks 1* | **0.396** | 0.626 | 0.532 | 0.863 | 0.397 |
| | *Blocks 2* | **0.406** | 0.618 | 0.546 | 0.878 | **0.406** |
| **Jena Weather 10T** | *Pointwise 1* | **0.086** | 0.629 | 0.132 | 0.710 | **0.086** |
| | *Pointwise 2* | **0.101** | 0.636 | 0.135 | 0.769 | **0.101** |
| | *Blocks 1* | **0.378** | 0.653 | 0.521 | 0.661 | 0.379 |
| | *Blocks 2* | **0.384** | 0.647 | 0.535 | 0.660 | **0.384** |
| **Jena Weather 1H** | *Pointwise 1* | **0.156** | 0.546 | 0.210 | 0.848 | **0.156** |
| | *Pointwise 2* | **0.214** | 0.628 | 0.260 | 0.906 | **0.214** |
| | *Blocks 1* | **0.366** | 0.544 | 0.517 | 0.791 | 0.367 |
| | *Blocks 2* | **0.370** | 0.538 | 0.525 | 0.821 | 0.371 |
| **Oikolab Weather** | *Pointwise 1* | **0.150** | 0.684 | 0.204 | 0.926 | 0.176 |
| | *Pointwise 2* | **0.228** | 0.765 | 0.263 | 0.967 | **0.228** |
| | *Blocks 1* | **0.449** | 0.681 | 0.605 | 0.886 | 0.450 |
| | *Blocks 2* | **0.454** | 0.683 | 0.611 | 0.909 | **0.454** |
| **PDB** | *Pointwise 1* | **0.062** | 0.345 | 0.207 | 1.020 | 0.063 |
| | *Pointwise 2* | **0.119** | 0.387 | 0.324 | 1.033 | **0.119** |
| | *Blocks 1* | **0.171** | 0.319 | 0.831 | 0.895 | **0.171** |
| | *Blocks 2* | **0.177** | 0.332 | 0.855 | 0.947 | **0.177** |
| **Pedestrian Counts** | *Pointwise 1* | **0.150** | 0.337 | 0.363 | 0.937 | 0.153 |
| | *Pointwise 2* | **0.200** | 0.369 | 0.443 | 0.970 | 0.214 |
| | *Blocks 1* | **0.172** | 0.327 | 0.863 | 0.820 | 0.174 |
| | *Blocks 2* | **0.178** | 0.327 | 0.880 | 0.846 | 0.181 |
| **Weather** | *Pointwise 1* | **0.257** | 0.655 | 0.273 | 0.915 | 0.259 |
| | *Pointwise 2* | **0.311** | 0.739 | 0.325 | 0.952 | 0.316 |
| | *Blocks 1* | **0.456** | 0.660 | 0.579 | 0.867 | 0.459 |
| | *Blocks 2* | **0.455** | 0.656 | 0.584 | 0.889 | **0.455** |
| **Mean (Std)** | | **0.252 (0.114)** | 0.554 (0.171) | 0.415 (0.244) | 0.904 (0.102) | 0.260 (0.116) |

**Results.** Table 10 shows a clear and consistent trend: the original feature set (`TabPFN1`) remains the strongest across virtually all datasets and settings. (i) Daily-only features (`TabPFN2`) perform substantially worse, highlighting the importance of capturing both daily and weekly periodicities. (ii) Using only the raw index (`TabPFN3`) yields reasonable but clearly inferior performance, especially in block-missing scenarios where periodic cues are most helpful. (iii) Random features degrade performance drastically (`TabPFN4`), confirming that `TabPFN-TS` is not invariant to arbitrary feature choice. (iv) Finally, augmenting correct features with random ones (`TabPFN5`) results in little to no degradation, indicating that the model is generally robust to irrelevant or noisy features and does not easily overfit to spurious temporal structure.

Overall, these results underline the importance of well-designed time features and suggest that part of `TabPFN-TS`'s effectiveness stems from the alignment between its meta-training distribution and the provided temporal encodings.

**Future work.** An important direction for future research is the automatic construction or selection of temporal features. Our experiments highlight that `TabPFN-TS` is highly sensitive to the structure of the

time encodings it receives. Instead of relying on manually crafted sinusoidal components, one could learn feature representations jointly with the model, or derive them through a separate meta-learning (like `MoTM`) or feature-search mechanism. Such an approach would be particularly valuable when dealing with highly heterogeneous datasets, spanning very different temporal granularities (minutes, hours, days, months) and periodic structures. Automatically discovering the most relevant periodicities and positional encodings would likely make `TabPFN-TS` more robust, more general, and less dependent on domain-specific feature engineering.

## D.2 Extended discussion on `TabPFN-TS` computational bottlenecks

A practical limitation of `TabPFN-TS` lies in its inference cost (see Section 4.1). On an NVIDIA H100 GPU, a single forward pass over a chunk of 672 time steps requires approximately one second. Depending on the computational resources available and the frequency at which predictions must be updated, this can be either acceptable or prohibitive. For example, batch offline imputation or low-frequency forecasting pipelines can easily amortize this cost, whereas real-time applications or systems operating on thousands of concurrent time series may find the latency challenging. Reducing inference time—through model distillation, sparse attention, chunk parallelization, or specialized kernels—thus represents a key avenue to make `TabPFN-TS` more broadly deployable in production environments.

In this regard, the recently introduced `TabPFN 2.5` (Grinsztajn et al., 2025) appears to be a promising solution, as it substantially accelerates in-context regression and could therefore mitigate the current inference bottlenecks of `TabPFN-TS`.

# E   Experiments with covariates: extensive qualitative results

## E.1   Qualitative results

To better visualize the impact of incorporating covariates, we present qualitative results for both `TabPFN-TS` and `MoTM`. Figure 10 and Figure 11 show examples from three datasets (*PV-France*, *Wind-France*, and *Load-France*), illustrating how the inclusion of covariate information helps each model reconstruct the four one-day missing blocks more accurately.

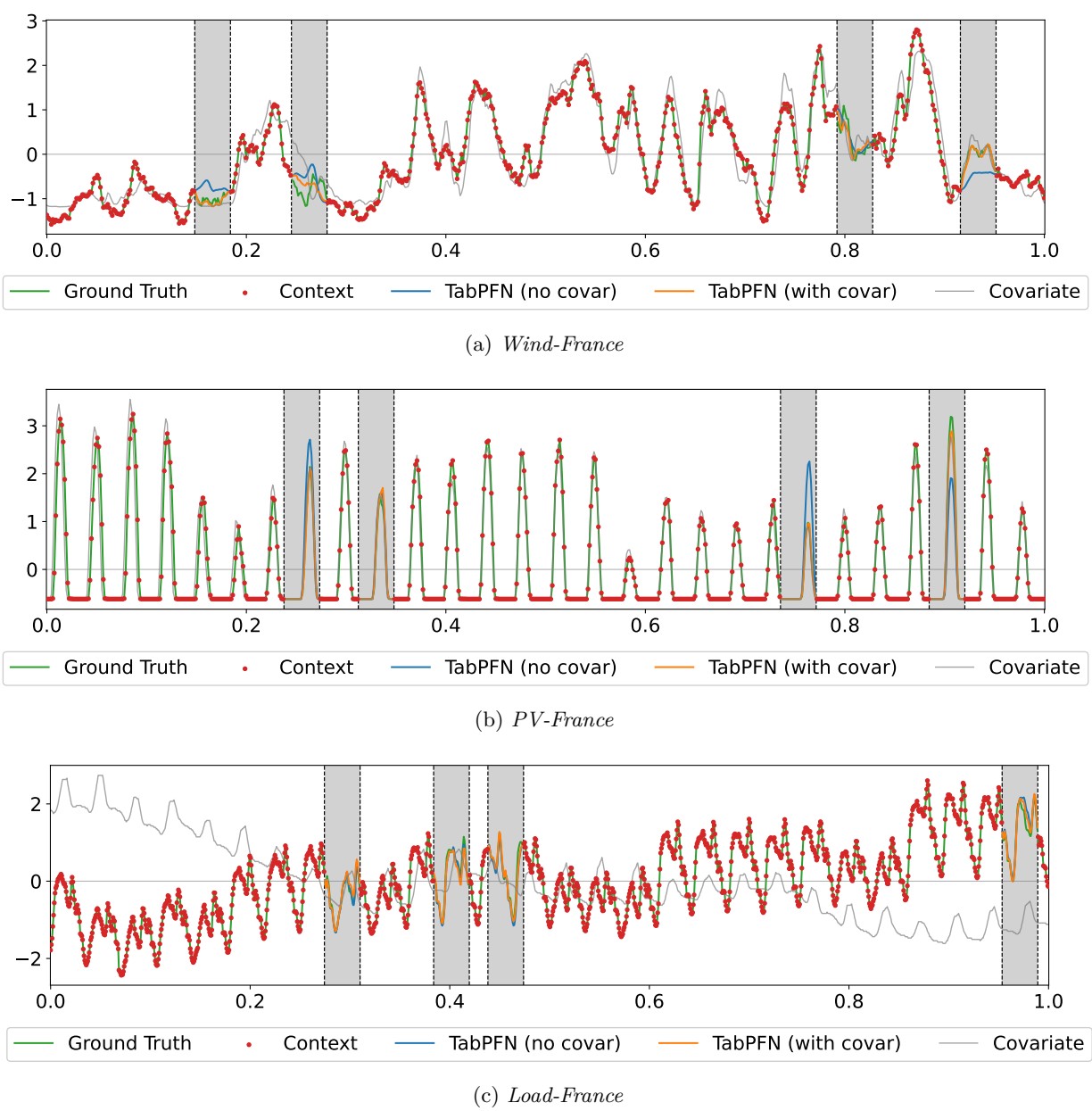

(a) *Wind-France*

(b) *PV-France*

(c) *Load-France*

Figure 10: `TabPFN-TS` qualitative results with and without covariates in the four one-day missing blocks scenario (Blocks 2).

**Results.**   As illustrated in Figure 10 and 11, incorporating covariates visually improves the reconstructions for both `TabPFN-TS` and `MoTM`. The covariate-enhanced versions capture missing intervals more smoothly and

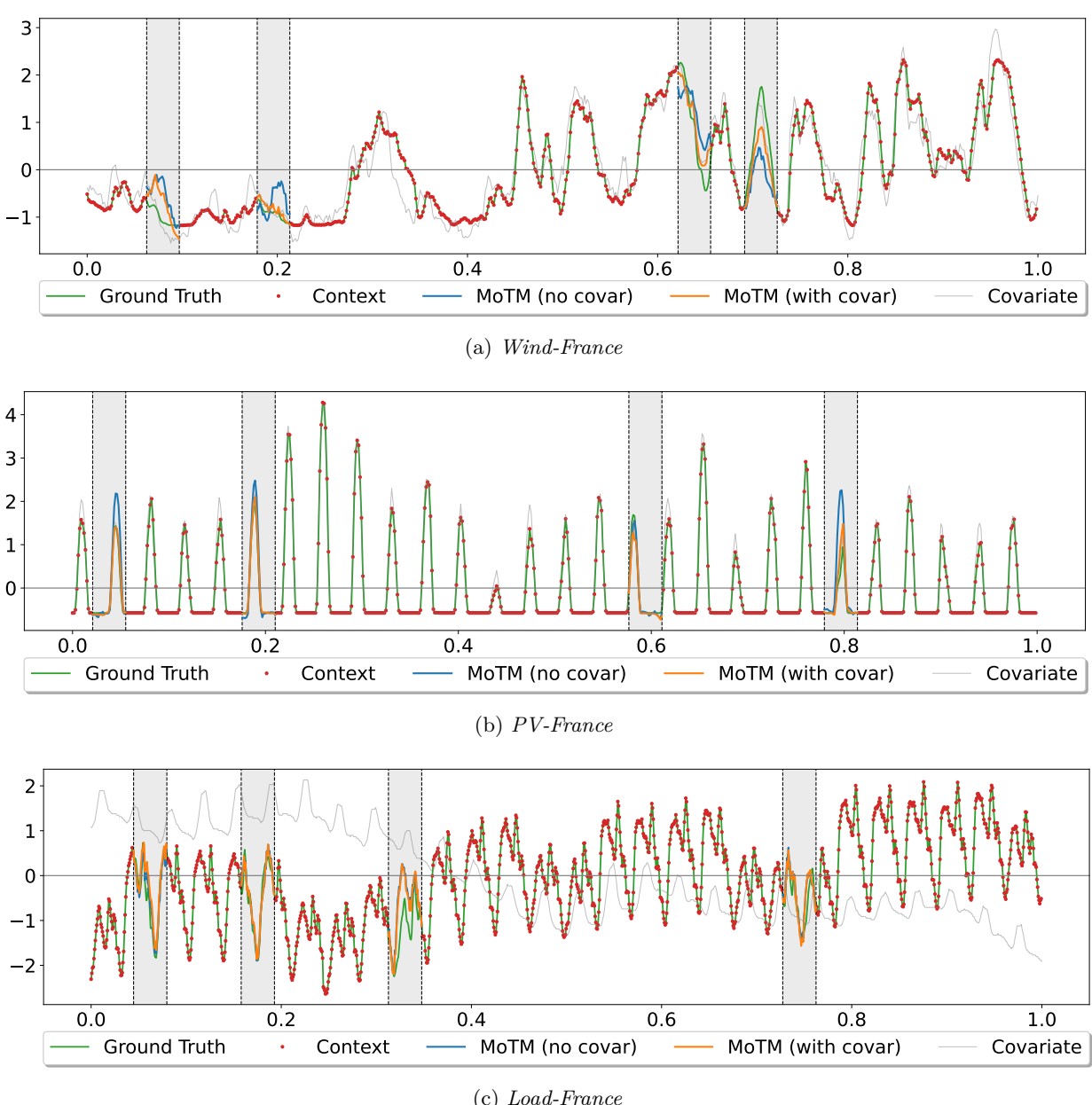

Figure 11: `MoTM` qualitative results with and without covariates in the four one-day missing blocks scenario (Blocks 2).

better follow short-term temporal variations within the four one-day gaps. We observe that the interpolations of `TabPFN-TS` align more closely with the ground truth and preserve sharper transitions across missing regions. In contrast, if `MoTM` consistently benefits from covariate information, its reconstructions sometimes struggle to capture sudden changes with great accuracy. Overall, the qualitative plots confirm that the most visible gains are observed on the *Wind-France* and *PV-France* datasets, while the impact remains limited for *Load-France*. One might note that quantile predictions can also be generated in this covariate setting, in a similar fashion to the univariate examples presented in Figure 11.

**Coarse granularity interpolation.** We extend our qualitative analysis to a practical scenario of interest, namely interpolating sparse yet regular observations, e.g. at a multi-hourly time-step, to a finer time

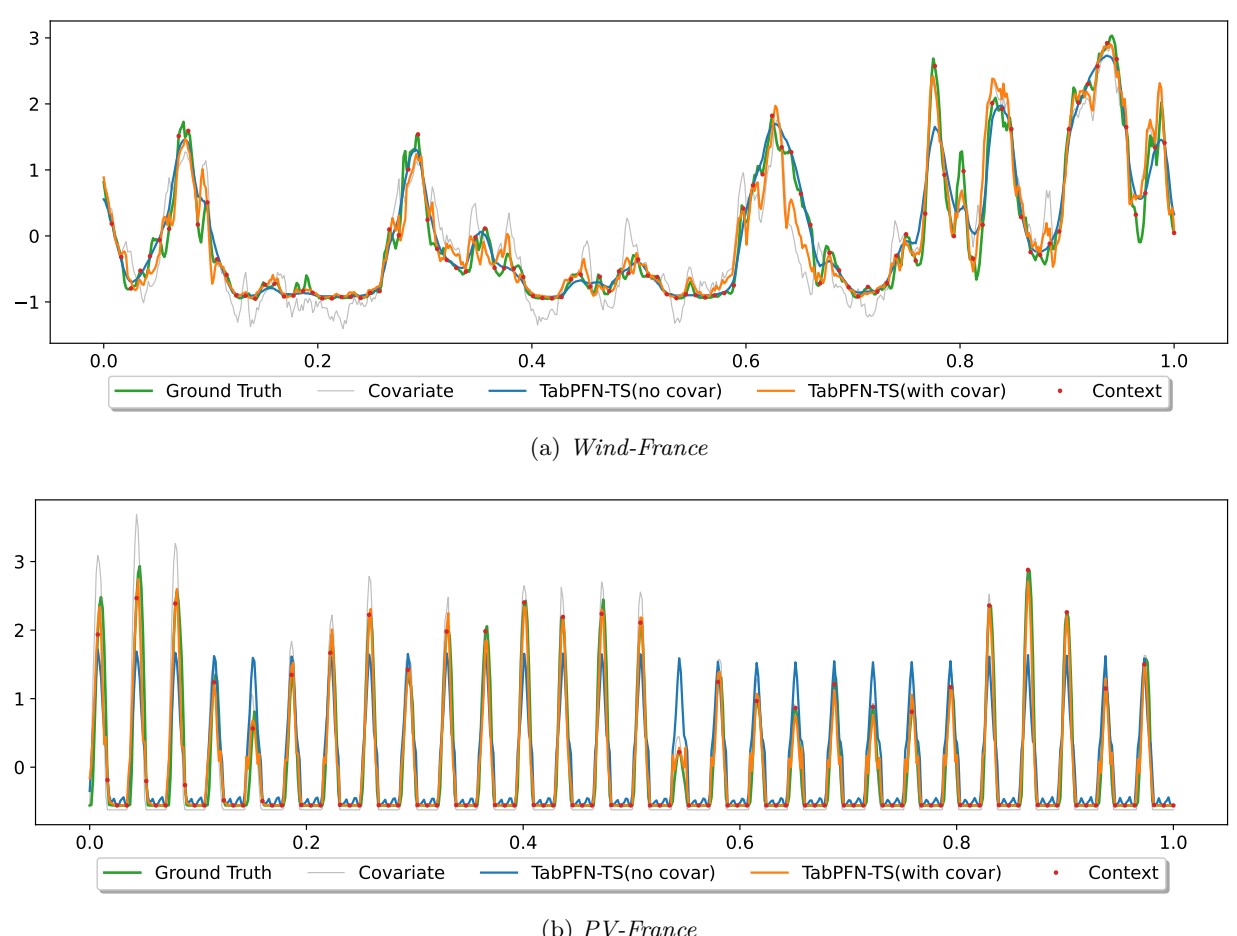

(a) *Wind-France*

(b) *PV-France*

Figure 12: `TabPFN-TS` qualitative results with and without covariates in coarse granularity scenarios: context observed at a six-hourly sampling rate, imputation on a dense hourly grid.

resolution, e.g. an hourly time-step. Figure 12 illustrates the behaviour of `TabPFN-TS` when interpolating the *Wind-France* and *PV-France* datasets from a six-hourly to an hourly sampling rate. On *PV-France*, we note that `TabPFN-TS` maintains shape consistency, although certain artifacts can be observed (typically non-zero values at night). These can be corrected by incorporating covariate information. On *Wind-France*, the model produces smooth univariate interpolations (blue line). Interestingly, adding the covariate is detrimental in this case: the in-context learning-based adaptation procedure of `TabPFN` on a sparse six-hour grid is mislead into spurious correlations by the stronger stochasticity of the covariate.

### E.2 Discussion: integration of incomplete covariate data

The integration mechanism used in Section 3.2 simply concatenates the covariate value $c(t)$ to the representation $H(t)$ before feeding it to the regression head $r_\theta(t)$. This assumes that the covariate is fully observed and temporally aligned with the target series for every timestamp $t$. When this assumption fails (for example, if the covariate contains missing values, is recorded asynchronously, or is sampled at a different frequency) naively integrating it into the model is not straightforward.

Below, we discuss practical strategies to handle such situations. These solutions are closer to implementation tricks than to principled methodological contributions, but they can be useful in practice.

**Partially observed but aligned covariates.** When the covariate is sampled on the same time grid as the target series but contains missing values, a straightforward approach is to first impute the covariate and then apply the standard integration procedure. This imputation can be performed using simple linear interpolation or more sophisticated approaches such as `TabPFN-TS` (or `MoTM` for faster imputation).

A limitation of this strategy is that the imputation of the covariate does not leverage the target series, which may lead to suboptimal reconstructions. More advanced approaches could jointly model both the target and the covariate — for example using implicit neural representations to obtain functional embeddings independent of the original sampling grid — but such ideas remain open research directions.

**Fully observed but misaligned covariates (asynchronous sensors or differing sampling rates).** If the covariate is complete but not recorded on the same timestamps as the target series, simple resampling methods such as linear interpolation can be used to align it with the target grid. More flexible approaches, such as querying `TabPFN-TS` or `MoTM` at arbitrary timestamps, can also be applied since these models operate in a continuous-time manner rather than imputing values through mask-based reconstruction (Tashiro et al., 2021; Du et al., 2023a). This makes them naturally suitable for harmonizing covariates recorded at heterogeneous frequencies.

## F  Comparison with hyperparameters-optimized `SAITS`

The experiments reported in the main paper rely on the hyperparameters recommended by each supervised baseline in their respective publications. While this ensures a fair and reproducible benchmark, it is also informative to compare `TabPFN-TS` and `MoTM` against a version of `SAITS` whose hyperparameters have been actively tuned. This comparison is particularly relevant given that hyperparameter optimization constitutes one of the main disadvantages of supervised approaches, in contrast with zero-shot methods.

**Setting.** We perform an explicit hyperparameter grid search for `SAITS`, varying its two main hyperparameters highlighted as important in the original paper, namely the inner model dimension ($d_{model}$) and the number of Transformer block layers ($n_{layers}$). The search is carried out on eleven representative datasets used throughout the benchmark—*BDG2-Bear, BDG2-Rat, Covid19 Energy, GFC12 Load, Hog, Jena Weather 10T, Jena Weather 1H, Oikolab Weather, PDB, Pedestrian Counts,* and *Weather*. For the hyperparameter search, the train/validation/test split follows a 0.7/0.1/0.2 ratio along the temporal axis, consistent with all other univariate experiments in the paper. For each dataset, we retain the hyperparameter configuration that yields the lowest validation MAE, averaged over the four validation-mask scenarios, and subsequently report the corresponding test MAE for each scenario. In addition, Figure 13 reports how the test MAE and training time vary with $d_{model}$ and $n_{layers}$ for `SAITS`.

Table 11: Best `SAITS` hyperparameter configuration per dataset based on validation MAE (averaged over the four validation-mask scenarios).

| Dataset | $d_{model}$ | $n_{layers}$ | Avg. Val. MAE |
|---|---|---|---|
| **BDG2-Bear** | 128 | 4 | 0.276 |
| **BDG2-Rat** | 128 | 3 | 0.328 |
| **Covid19 Energy** | 128 | 3 | 0.213 |
| **GFC12 Load** | 128 | 3 | 0.289 |
| **Hog** | 128 | 4 | 0.416 |
| **Jena Weather 10T** | 256 | 2 | 0.367 |
| **Jena Weather 1H** | 128 | 4 | 0.390 |
| **Oikolab Weather** | 128 | 3 | 0.430 |
| **PDB** | 256 | 3 | 0.249 |
| **Pedestrian Counts** | 256 | 4 | 0.165 |
| **Weather** | 256 | 2 | 0.412 |

Table 12: Comparison of test MAE between the two zero-shot approaches, `TabPFN-TS` and `MoTM`, and the tuned `SAITS` baseline (best configuration selected per dataset via validation MAE). For each setting, the best (lowest) MAE is shown in **bold**, and the second best is underlined.

| Dataset | Setting | TabPFN-TS | MoTM | SAITS$_{opt}$ | SAITS$_{base}$ |
|---|---|---|---|---|---|
| **BDG2–Bear** | *Pointwise 1* | **0.171** | 0.202 | 0.191 | 0.241 |
| | *Pointwise 2* | **0.223** | 0.240 | 0.253 | 0.309 |
| | *Blocks 1* | **0.272** | 0.332 | 0.327 | 0.399 |
| | *Blocks 2* | **0.280** | 0.336 | 0.325 | 0.405 |
| **BDG2–Rat** | *Pointwise 1* | **0.196** | 0.231 | 0.199 | 0.266 |
| | *Pointwise 2* | **0.256** | 0.273 | 0.270 | 0.339 |
| | *Blocks 1* | **0.349** | 0.400 | 0.421 | 0.495 |
| | *Blocks 2* | **0.355** | 0.402 | 0.420 | 0.497 |
| **Covid19 Energy** | *Pointwise 1* | **0.075** | 0.099 | 0.344 | 0.399 |
| | *Pointwise 2* | 0.132 | **0.127** | 0.380 | 0.417 |
| | *Blocks 1* | **0.202** | 0.222 | 0.372 | 0.432 |
| | *Blocks 2* | **0.201** | 0.232 | 0.367 | 0.436 |
| **GFC12 Load** | *Pointwise 1* | **0.143** | 0.189 | 0.189 | 0.188 |
| | *Pointwise 2* | 0.237 | **0.231** | 0.277 | 0.280 |
| | *Blocks 1* | **0.353** | 0.382 | 0.399 | 0.417 |
| | *Blocks 2* | **0.363** | 0.387 | 0.406 | 0.425 |
| **Hog** | *Pointwise 1* | **0.196** | 0.240 | 0.226 | 0.245 |
| | *Pointwise 2* | **0.260** | 0.286 | 0.292 | 0.320 |
| | *Blocks 1* | **0.396** | 0.458 | 0.514 | 0.518 |
| | *Blocks 2* | **0.406** | 0.464 | 0.516 | 0.528 |
| **Jena Weather 10T** | *Pointwise 1* | **0.086** | 0.190 | 0.241 | 0.177 |
| | *Pointwise 2* | **0.101** | 0.207 | 0.293 | 0.196 |
| | *Blocks 1* | **0.378** | 0.453 | 0.502 | 0.557 |
| | *Blocks 2* | **0.384** | 0.459 | 0.504 | 0.561 |
| **Jena Weather 1H** | *Pointwise 1* | **0.156** | 0.224 | 0.241 | 0.346 |
| | *Pointwise 2* | **0.214** | 0.272 | 0.293 | 0.394 |
| | *Blocks 1* | **0.366** | 0.463 | 0.502 | 0.532 |
| | *Blocks 2* | **0.370** | 0.464 | 0.504 | 0.532 |
| **Oikolab Weather** | *Pointwise 1* | **0.150** | 0.227 | 0.188 | 0.207 |
| | *Pointwise 2* | **0.228** | 0.278 | 0.293 | 0.294 |
| | *Blocks 1* | **0.449** | 0.529 | 0.598 | 0.581 |
| | *Blocks 2* | **0.454** | 0.534 | 0.602 | 0.584 |
| **PDB** | *Pointwise 1* | **0.062** | 0.094 | 0.201 | 0.337 |
| | *Pointwise 2* | **0.119** | 0.121 | 0.271 | 0.373 |
| | *Blocks 1* | **0.171** | 0.197 | 0.317 | 0.353 |
| | *Blocks 2* | **0.177** | 0.201 | 0.314 | 0.373 |
| **Pedestrian Counts** | *Pointwise 1* | **0.150** | 0.196 | 0.128 | 0.240 |
| | *Pointwise 2* | **0.200** | 0.239 | 0.183 | 0.289 |
| | *Blocks 1* | **0.172** | 0.254 | 0.173 | 0.243 |
| | *Blocks 2* | **0.178** | 0.260 | 0.176 | 0.252 |
| **Weather** | *Pointwise 1* | **0.257** | 0.298 | 0.281 | 0.290 |
| | *Pointwise 2* | **0.311** | 0.342 | 0.348 | 0.364 |
| | *Blocks 1* | **0.456** | 0.494 | 0.556 | 0.592 |
| | *Blocks 2* | **0.455** | 0.488 | 0.553 | 0.590 |
| **Mean (Std)** | | **0.252 (0.114)** | 0.300 (0.122) | 0.340 (0.159) | 0.379 (0.122) |

**Results.** Table 12 reports the detailed test MAE for each dataset and masking scenario. SAITS$_{opts}$ correspond to the best configurtaion and SAITS$_{base}$ correspond to the default hyperparameters baseline provided by the authors (see Section 2.1). Table 11 report the best configuration for each datasets according to the lowest average MAE obtained on the validation set. Overall, hyperparameter tuning yields only modest gains for `SAITS`: the tuned configuration slightly improves over the baseline, but is still outperformed, on average across the 11 datasets, by the two zero-shot approaches `TabPFN-TS` and `MoTM` by an average of 26%

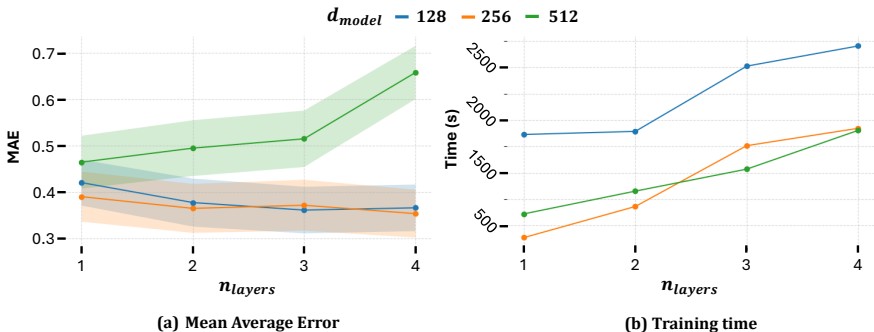

Figure 13: Hyperparameter ablation for SAITS. (a) Average test MAE across the 11 datasets as a function of the number of layers, with one curve per $d_{model}$. (b) Average training time across the 11 datasets as a function of the number of layers, again with one curve per $d_{model}$.

and 12% respectively. In addition, Figure 13(a) shows that, for $d_{model} \in \{128, 256\}$ and $n_{layers} \in \{2, 3, 4\}$, the average test MAE remains essentially unchanged, indicating limited benefits from increasing depth within this range. By contrast, Figure 13(b) highlights that increasing either $d_{model}$ or $n_{layers}$ leads to a substantial increase in training time. These findings corroborate the default configuration proposed by the authors of SAITS (Du et al., 2023a), namely $d_{model} = 256$ and $n_{layers} = 2$.

