# OpenReview forum: "Are Time-Indexed Foundation Models the Future of Time Series Imputation?"
_TMLR — Accepted by TMLR_

### Review · Reviewer_e9qb · 2025-11-17

**Summary Of Contributions:**

This paper presents a large-scale empirical evaluation of time-indexed foundation models, specifically TabPFN-TS and MoTM, for zero-shot time-series imputation. The authors benchmark these models across 33 out-of-domain datasets under several missingness regimes and show that they consistently outperform both classical local imputers and state-of-the-art supervised deep-learning baselines. The paper additionally evaluates the ability of these models to integrate covariates at inference time without retraining, and provides uncertainty-quantification results as well as computational analyses

Strengths:
- Clear problem motivation: Imputation in heterogeneous, irregular time series is an important and underexplored area compared to zero-shot forecasting.
- Strong empirical work: The benchmark is large, diverse, and carefully designed to avoid leakage from MoTM pretraining datasets.
- Comprehensive comparisons: Includes local, supervised, and foundation baselines, with a transparent discussion of why some were excluded.
- Covariate integration analysis is thorough and practically relevant, especially for domains such as energy and climate.
- Uncertainty quantification experiments are well executed and strengthen the argument for real-world applicability.

Weaknesses:
- Limited practical validation for physical or dynamical constraints: While extensive, the evaluation focuses mainly on statistical accuracy. For domains like energy, weather, or PV power, shape constraints (monotonicity, smoothness, daily cycles) matter and are not examined.
- Interpretability of failure modes: The paper shows strong average performance, but provides little discussion of when these models behave pathologically (oversmoothing, unrealistic extrapolation, seasonal misalignment).
- Computational cost discussion is descriptive but not actionable: The results are clear, but the paper does not thoroughly analyze why TabPFN-TS becomes slow at scale or how its context length limits affect performance. Especially for energy data, the applicability of imputation models for trading or scheduling purposes largely relies on inference time.
- Covariate integration is tested only with fully observed covariates; some important use-cases involve sparse or noisy covariates.
- Real-world scenarios such as reduced temporal granularity (e.g., 6-hourly forecast horizons) are not studied but highly relevant for the types of time-indexed interpolations the paper discusses. Certain practical claims (e.g., applicability to real-world industrial time-series settings) would benefit from additional qualitative analysis of shape fidelity and physical consistency..

**Additional Comments:**

- For the PV-France dataset, are block-missing imputations also visually coherent from a physical standpoint (e.g., smooth bell-shaped curves, monotonic morning ramps unless cloud-driven deviations)?
- Do the authors have insight into how sensitive these models are to the choice of Fourier frequencies in TabPFN-TS, especially for datasets with irregular seasonality (e.g., traffic or weather anomalies)?

**Audience:**

Yes

**Audience Explanation:**

Time-series foundation models are gaining increasing attention, yet the literature is heavily skewed toward forecasting rather than imputation. A systematic evaluation of zero-shot imputation is timely and valuable for researchers working on foundation models, representation learning, and applied ML in domains such as manufacturing, energy, and sensor networks. The covariate-integration results, in particular, highlight a capability that is widely demanded in practice and rarely addressed in zero-shot settings.

**Broader Impact Concerns:**

No concern.

**Claims And Evidence:**

Yes

**Claims Explanation:**

The empirical evidence supports the main claims convincingly. The superiority of TabPFN-TS and MoTM over both supervised and local models is consistently demonstrated across datasets, missingness patterns, and covariate settings. Method descriptions are sufficiently detailed to understand why zero-shot generalization is expected.

**Requested Changes:**

- Physical/shape consistency analysis for energy and climate datasets. For instance, solar power or irradiance curves exhibit known smoothness and monotonic patterns driven by diurnal cycles. It would be useful to evaluate whether the imputations produced by foundation models adhere to physically sensible shapes or occasionally produce unrealistic artifacts. This is particularly relevant for the PV-France case.
- Discussion of reduced-granularity interpolation use cases. Many operational datasets (e.g., ECMWF IFS/AIFS) provide high granularity in the near term but only 3–6-hourly values for longer horizons. Could the proposed models interpolate such coarse weather forecasts into hourly profiles? Even a short exploratory discussion would improve practical relevance.
- Failure-mode characterization. Given the scale of the benchmark, qualitative cases of poor or unstable reconstructions would help users understand when the models should not be deployed.
- Clarify limitations of covariate integration. The experiments assume fully observed covariates. A short discussion of situations where covariates are missing, partially observed, or noisy would be valuable.
- More actionable discussion on computational bottlenecks. TabPFN-TS is extremely slow for large contexts; explaining the drivers and possible mitigation strategies would be useful.

---

> ### Author Response · Authors · 2025-12-11
> **Answer (1/2)**
>
> We sincerely thank the reviewer for the time spent on this review. We appreciate the level of understanding and details in the review. We hope to address the points raised below.
>
> ## Weaknesses
>
> > **Weakness 1.** Limited practical validation for physical or dynamical constraints: While extensive, the evaluation focuses mainly on statistical accuracy. For domains like energy, weather, or PV power, shape constraints (monotonicity, smoothness, daily cycles) matter and are not examined.
>
> **Answer Weakness 1.**
> Please see our answers to Weakness 2 and RC1.
>
> > **Weakness 2.** Interpretability of failure modes: The paper shows strong average performance, but provides little discussion of when these models behave pathologically (oversmoothing, unrealistic extrapolation, seasonal misalignment).
>
> **Answer Weakness 2.**
> We thank the reviewers for highlighting these practical considerations. Indeed, TabPFN-TS achieves strong performance across datasets and settings, which can make it difficult to pinpoint specific failure modes or dataset-dependent guidance. To address this, we added Appendix D.2 to provides more actionable guidance for practitioners regarding both performance and practical deployment trade-offs.
>
> > **Weakness 3.** Computational cost discussion is descriptive but not actionable: The results are clear, but the paper does not thoroughly analyze why TabPFN-TS becomes slow at scale or how its context length limits affect performance. Especially for energy data, the applicability of imputation models for trading or scheduling purposes largely relies on inference time.
>
> **Answer Weakness 3.**
> Thank you for this recommendation; indeed, we have added a new dedicated appendix (Appendix D.2) to address the computational limitations of TabPFN-TS. As detailed there, the main bottleneck arises from inference latency: processing a 672-step chunk takes roughly one second on an NVIDIA H100. While this is acceptable for offline or low-frequency forecasting, it can become restrictive for real-time applications or large-scale scenarios involving many concurrent time series. Appendix D.2 also discusses the recently introduced TabPFN 2.5, which substantially accelerates in-context regression and could therefore help alleviate the current computational bottlenecks of TabPFN-TS.
>
> > **Weakness 4.** Covariate integration is tested only with fully observed covariates; some important use-cases involve sparse or noisy covariates.
>
> **Answer Weakness 4.**
> Thank you for the suggestion; we have added a dedicated appendix (Appendix E.2) to discuss this limitation. As explained there, our current covariate integration mechanism assumes fully observed and aligned covariates, which does not cover some important real-world cases. Appendix E.2 summarizes practical strategies for handling incomplete or misaligned covariates — including simple or learned imputation strategies, resampling, and the use of continuous-time models such as TabPFN-TS or MoTM to harmonize heterogeneous sampling rates.
>
> > **Weakness 5.** Real-world scenarios such as reduced temporal granularity (e.g., 6-hourly forecast horizons) are not studied but highly relevant for the types of time-indexed interpolations the paper discusses. Certain practical claims (e.g., applicability to real-world industrial time-series settings) would benefit from additional qualitative analysis of shape fidelity and physical consistency..
>
> **Answer Weakness 5.**
> We thank the reviewer for pointing to this challenging real-world scenario. We conducted in Appendix E.1 additional qualitative analysis on PV-France and Wind-France to interpolate from a sparse six-hour time grid to an hourly time resolution. In the univariate setting, TabPFN-TS does maintain shape consistency on PV (bell-shape curves, daily cycles, etc.) but introduces small artifacts (non-zeros values at night) that were absent from our experiments with sparse missing at random observations.
> On Wind-France, the interpolation is smooth and shows high fidelity to the ground truth signal.
>
> Adding the covariate information (fully observed on the hourly grid) produces distinct effects.
> On PV-France, it clearly helps adjusting the univariate interpolation (adjusted peak values, zero-values at night).
> On the other hand, on Wind-France, adding the covariate is detrimental to the target interpolation.
> We hypothesize that this is caused by (1) the strong stochasticity of the covariate that (2) misleads the adaptation mechanism of TabPFN (in-context-learning on the six-hourly grid) into spurious correlations.
>
> This discussion is included at the end of Appendix E.1 in the revised version of the manuscript.

---

> ### Author Response · Authors · 2025-12-11
> **Answer (2/2)**
>
> ## Requested Changes
>
> > **Requested Change 1.** Physical/shape consistency analysis for energy and climate datasets. For instance, solar power or irradiance curves exhibit known smoothness and monotonic patterns driven by diurnal cycles. It would be useful to evaluate whether the imputations produced by foundation models adhere to physically sensible shapes or occasionally produce unrealistic artifacts. This is particularly relevant for the PV-France case.
>
> **Answer Requested Change 1.**
> Thank you for this question. Visually, on the PV-France dataset (see Figures 10.b and 12.b), we observe that TabPFN-TS behaves consistently with physical expectations: it imputes strictly zero values at night and reproduces the typical bell-shaped curves of PV curves. In contrast, MoTM occasionally imputes small non-zero values during nighttime (see Figure 11.b), which is physically inconsistent and highlights an important area for improvement for this model.
>
> > **All remaining requested changes have already been addressed in the Weaknesses section. We thank you again for your very helpful comments and for taking the time to review our work.**
>
> ## Additional Comments
>
> > **Additional Comment 1.**
> For the PV-France dataset, are block-missing imputations also visually coherent from a physical standpoint (e.g., smooth bell-shaped curves, monotonic morning ramps unless cloud-driven deviations)?
>
> **Answer Additional Comment 1.** Please see our answer to RC 1
>
> > **Additional Comment 2.**
> Do the authors have insight into how sensitive these models are to the choice of Fourier frequencies in TabPFN-TS, especially for datasets with irregular seasonality (e.g., traffic or weather anomalies)?
>
> **Answer Additional Comment 2.**
> Thank you for this insightful suggestion. We have added Appendix D.1, which provides a dedicated analysis of TabPFN-TS’s sensitivity to the choice of Fourier features. In short, we systematically vary the time encodings across 11 representative datasets and observe that the original feature design (daily + weekly periodicities together with the raw time index) consistently yields the strongest zero-shot performance. Removing relevant frequencies or replacing them with random ones leads to clear degradation, while adding spurious features has limited impact. Overall, the results confirm that well-chosen temporal encodings matter and Appendix D.1 now documents this behavior in detail.

---

### Review · Reviewer_ixDg · 2025-11-17

**Summary Of Contributions:**

The paper conducts a study of time-indexed time series foundation models as zero-shot imputers that do not require training, by comparing them to a variety of models that do require training. The main thesis of the paper is that that modern foundation models are good zero-shot imputers, and this thesis is corroborated by a large set of experiments. The paper does not propose a new theory, algorithm, or a learning methodology - but rather proves a position by an extensive set of empirical experiments.

**Audience:**

Yes

**Audience Explanation:**

I do see a lot of value in such a paper by directing practitioners and researchers to default "well-working" methods.

**Claims And Evidence:**

No

**Claims Explanation:**

Since the paper's main thesis is corroborated mainly empirically, the evidence would be more convincing if:
1. It was clear exactly how the non zero-shot models were trained. Pointing to a library is not enough - each combination of a model and a data-set may require a different set of training parameters, such as a learning rate, weight decay, etc.. It may be the case that some models under-perform simply because they were trained with unfit parameters. This, of course, can also be a caveat of these models - they require tuning, but this should be clearly articulated in the paper. For example, a result with thorough tuning and another one without, at least for **one** of the models, for which the computational cost is not that large, would be more convincing (unless everything is tuned - I don't know, it's not clear from the paper).
2. It would be more convincing if it was easily reproducible. I do not see a pointer to an (anonymized) code repo for reproducing the results. Pointing to an external library is not enough. There are other reproducibility concerns, such as the ones above. Code would remove all doubt.

**Requested Changes:**

Please see the explanation for why the evidence is not convincing - address the tuning concern in some convincing way, or provide code.

---

> ### Author Response · Authors · 2025-12-11
> **Answer**
>
> We thank the reviewer for their careful review and valuable feedback. We address the points below.
>
> ## Weaknesses
>
> > **Weakness 1.** It was clear exactly how the non zero-shot models were trained. Pointing to a library is not enough - each combination of a model and a data-set may require a different set of training parameters, such as a learning rate, weight decay, etc.. It may be the case that some models under-perform simply because they were trained with unfit parameters. This, of course, can also be a caveat of these models - they require tuning, but this should be clearly articulated in the paper. For example, a result with thorough tuning and another one without, at least for one of the models, for which the computational cost is not that large, would be more convincing (unless everything is tuned - I don't know, it's not clear from the paper).
>
> **Answer Weakness 1.** Thank you very much for raising this important point, and we apologize that the initial submission did not make this sufficiently clear.
> We have substantially reorganized Appendix A.2 to clarify our training protocol for all supervised baselines. In summary, each supervised model is trained using the hyperparameter configuration recommended in its respective original paper.
> In addition, to avoid overfitting and to ensure that all models are trained as fairly as possible, we used a spare validation split for every dataset and apply early stopping based on validation performance.
>
> Regarding your second concern, namely that some supervised baselines might underperform due to suboptimal hyperparameters, we have added a new Appendix F where we perform an explicit hyperparameter search for SAITS, the strongest supervised baseline in our benchmark.
> We consider eleven representative datasets and systematically vary the two main hyperparameters emphasized in the original SAITS paper, the inner model dimension $d_{model}$ and the number of Transformer blocks $n_{layers}$.
> For each dataset, we select the configuration that achieves the lowest validation MAE (averaged over the four validation-mask scenarios) and then report the corresponding test MAE per scenario (Table 11  and Table 12). The results show that hyperparameter tuning yields only modest gains for SAITS: the tuned configuration slightly improves over the default one, but, on average across the 11 datasets, it is still outperformed by the two zero-shot approaches (TabPFN-TS and MoTM) by more than 12%, as reported in Table 12.
> Furthermore, the ablation in Figure 12 shows that, for $d_{model} \in \{128, 256\}$ and $n_{layers} \in \{2, 3, 4\}$, the average test MAE remains essentially unchanged, whereas increasing either $d_{model}$ or $n_{layers}$ beyond this range leads to a substantial increase in training time without commensurate performance gains.
> These findings corroborate the default configuration proposed by the SAITS authors ($d_{model}=256$, $n_{layers}=2$) and indicate that our main conclusions are not driven by an unfavorable choice of hyperparameters for this baseline.
> In addition, this hyperparameter search is computationally expensive, as it requires training a separate model for each configuration.
> We now explicitly state in the paper that supervised models may require additional tuning, but that, at least for SAITS, such tuning does not close the performance gap with the proposed zero-shot methods.
>
> > **Weakness 2.** It would be more convincing if it was easily reproducible. I do not see a pointer to an (anonymized) code repo for reproducing the results. Pointing to an external library is not enough. There are other reproducibility concerns, such as the ones above. Code would remove all doubt.
>
> Thank you for emphasizing the importance of reproducibility. We fully agree. An anonymized code repository is now provided at <https://anonymous.4open.science/r/tmlr-submission-timeindexed-tsfm/>. It contains all scripts needed to understand the paper experiments pipeline.
>
> ## Requested Changes
>
> > Please see the explanation for why the evidence is not convincing - address the tuning concern in some convincing way, or provide code.
>
> **Answer.** Please see our answers to the Weaknesses.

---

> > ### Comment · Reviewer_ixDg · 2025-12-16
> >
> > Thank you for your answers. Your changes make the evidence more convincing. I also took a look at the code - nicely structured and well-documented enough to base future research or experiments on.

---

> > > ### Author Response · Authors · 2025-12-16
> > > **Answer**
> > >
> > > We thank the reviewer for their valuable comments and are pleased that the provided code was found to be useful.

---

### Review · Reviewer_nTQe · 2025-11-27

**Summary Of Contributions:**

**Summary** - The paper presents an empirical study of time-indexed foundation models for zero-shot time-series imputation. The work compares two time-indexed foundation models with supervised and local imputation models. The evaluation is performed under four different settings - point-wise masking (50% and 70%) and block-wise masking (two or four entire days masking) across 33 univariate out-of-distribution datasets.

**Strengths**
1. The paper conducts extensive evaluation on 33 OOD datasets
2. Several qualitative results are included and analysed.
3. Discusses the computational cost along with the empirical performance

**Weaknesses**
1. There are other Time-series Foundation Models like MOMENT, Frozen Pretrained Transformer (FPT), NuwaTS, that claim time-series imputation ability. Given the primary focus of the work is on time-series foundation models, it is important to include all the relevant foundation models for a broader evaluation benchmark. While the paper acknowledges the limitations of MOMENT for the current evaluation, it would be good to report the results. It would be informative for the community interested in TSFMs for imputation.
2. Related works section is too long. The paper spends a significant amount of text detailing the internals of the time-indexed foundation models - MoTM and TabPFN-TS (Section 2.2 and Appendix A.3 repeats much of the architectural description)
3. The tables do not report standard deviation, which makes it difficult to determine if the reported gains are statistically significant (for e.g. in Table 2, the numbers are very close, std would help clarify significance of the gain)
4. On a high-level, the work simply evaluates imputation models on different datasets and reports the results. It does not really provide any insights based on the results

**Audience:**

Yes

**Audience Explanation:**

The paper explores the rapidly emerging Time series Foundation Models, for time series imputation, which is an active field of research, with high interest across multiple domains (healthcare, climate, industry).

**Broader Impact Concerns:**

No major concerns on the ethical or societal implications of the work.

**Claims And Evidence:**

Yes

**Claims Explanation:**

Most of the claims are well-supported by the experiments. TabPFN-TS and MoTM consistently achieve lowest MAE and best average ranks across benchmark datasets, and across all four missingness settings. However, the following claims are slightly less convincing and require more evidence,
(a) integration of covariates lead to significant results - without multiple trial runs, it is difficult to assess the significance of the gains
(b) the evaluation is extensive - not all relevant foundation models are included in the comparison

**Requested Changes:**

1. Consider other Foundation Models into the evaluation framework - MOMENT, Frozen Pretrained Transformer (FPT), NuwaTS. It would be helpful to report the results of MOMENT, with and without fine-tuning (Critical)
2. Results reported should be based on multiple trials, and should include the standard deviation (Critical)
3. Provide more reasoning behind the observed results, for e.g. (a) why do we see CSDI or TimesNet performing poorly on the OOD Datasets (and also in terms of the average rank), or why BRITS or even the local baselines have a better rank than CSDI/TimesNet - is it because of the size of training dataset, or does it require more hyper-parameter tuning. (Strengthening)
4. Given linear interpolation shows competitive performance on pointwise setting, it would be useful to add a simple non-linear interpolator like Spline into the evaluation benchmark as well. (Strengthening)
5. Related works section can be made more concise, as the Appendix covers the architectures of the foundation models in detail. This will improve the readability of the paper. (Strengthening)
6. Currently, the covariates considered are fully observed. However, in real-world multivariate setting, multiple variables often have concurrent missing values. It would be useful to add a discussion or empirical results by carrying out experiments to show the effect of partially observed covariates (Strengthening)

---

> ### Author Response · Authors · 2025-12-11
> **Answer (1/2)**
>
> We thank the reviewer for their thorough evaluation and valuable feedback.
> We appreciate the detailed insights provided and address the comments below.
>
> ## Weaknesses
>
> > **Weakness 1.** There are other Time-series Foundation Models like MOMENT, Frozen Pretrained Transformer (FPT), NuwaTS, that claim time-series imputation ability. Given the primary focus of the work is on time-series foundation models, it is important to include all the relevant foundation models for a broader evaluation benchmark. While the paper acknowledges the limitations of MOMENT for the current evaluation, it would be good to report the results. It would be informative for the community interested in TSFMs for imputation.
>
>
> **Answer weakness 1.** Thank you for this very relevant question. We have significantly expanded Appendix A.4 (“Excluded baselines”) to discuss additional TSFMs not included in the main benchmark, namely MOMENT and NuwaTS. Since MOMENT is a successor to FPT, we only include MOMENT as representative of this family.
>
> We also conducted a quantitative assessment (Appendix A.4.1) of MOMENT, NuwaTS, and Cubic Spline interpolation on eleven representative datasets.
> MOMENT underperforms across all datasets, reflecting its design for fine-tuning rather than zero-shot imputation.
> NuwaTS does better than MOMENT but still lags substantially behind TabPFN-TS and MoTM.
> Cubic Spline interpolation performs worse than simple linear interpolation, especially under block missingness.
>
> These results confirm that time-index foundation models (TabPFN-TS and MoTM) remain by far the strongest zero-shot models. Fine-tuning MOMENT could improve its performance, but doing so would require fine-tuning all foundation models, which is outside the scope of this work that focuses on zero-shot capabilities.
>
> > **Weakness 2.** Related works section is too long. The paper spends a significant amount of text detailing the internals of the time-indexed foundation models - MoTM and TabPFN-TS (Section 2.2 and Appendix A.3 repeats much of the architectural description)
>
> **Answer weakness 2.** Thank you for this comment. We have revised the paper to reduce repetitions and make the presentation of the time-indexed foundation models (MoTM and TabPFN-TS) clearer.
>
> > **Weakness 3.** The tables do not report standard deviation, which makes it difficult to determine if the reported gains are statistically significant (for e.g. in Table 2, the numbers are very close, std would help clarify significance of the gain)
> On a high-level, the work simply evaluates imputation models on different datasets and reports the results. It does not really provide any insights based on the results
>
> **Answer weakness 3.** Thank you for this comment. For zero-shot foundation models, which are deterministic, as well as for local baselines, the results are always identical, so standard deviations across runs are not meaningful. To provide a measure of variability, we have revised all univariate results tables to report the mean and the standard deviation calculated across the datasets. This quantifies the robustness and generalization capability across various time series.
>
> Regarding the supervised baselines performance variations, we introduce Appendix F (Comparison with hyperparameters-optimized SAITS) in the revised manuscript. This section shows that even with hyperparameters optimization on eleven datasets, the performance gain for SAITS is marginal, and it consistently remains behind the zero-shot models (on these datasets).

---

> ### Author Response · Authors · 2025-12-11
> **Answer (2/2)**
>
> ## Requested changes
>
> > **Requested change 3.** Provide more reasoning behind the observed results, for e.g. (a) why do we see CSDI or TimesNet performing poorly on the OOD Datasets (and also in terms of the average rank), or why BRITS or even the local baselines have a better rank than CSDI/TimesNet - is it because of the size of training dataset, or does it require more hyper-parameter tuning. (Strengthening)}
>
> **Answer requested change 3.** We thank the reviewer for this insightful comment and agree that the behaviour of CSDI/TimesNet compared to BRITS and the local baselines deserves further discussion.
> By cross-referencing the full OoD results in Table 6 (Appendix C.1) with the number of training samples reported in Table 4 (Appendix B.1), we observe that many of the OoD datasets correspond to a small-data regime, with only a few hundred (or fewer) series available for training.
>
> In this setting, architectures such as CSDI and TimesNet appear to be particularly ill-suited and seems to require a large number of training examples to generalize.
> In contrast, BRITS has a lighter recurrent architecture with stronger inductive biases, and the local baselines do not train at all; both are therefore more robust in the small-data regime, which likely explains their better average rank.
>
> This interpretation is further supported by the experiments in Appendix C.3, where we study  CSDI on lower-frequency datasets. On the smallest datasets in our benchmark (e.g., Births Daily and Births Weekly, which each contain a single univariate series of length ≈ 1000 and ≈ 3000), CSDI performs particularly poorly.
>
> > **Requested change 6.** Currently, the covariates considered are fully observed. However, in real-world multivariate setting, multiple variables often have concurrent missing values. It would be useful to add a discussion or empirical results by carrying out experiments to show the effect of partially observed covariates (Strengthening)
>
> **Answer requested change 6.** Thank you for the suggestion; we have added a dedicated appendix (Appendix E.2) to discuss this limitation. As explained there, our current covariate integration mechanism assumes fully observed and aligned covariates, which does not cover some important real-world cases. Appendix E.2 summarizes practical strategies for handling incomplete or misaligned covariates — including simple or learned imputation strategies, resampling, and the use of continuous-time models such as TabPFN-TS or MoTM to harmonize heterogeneous sampling rates.
>
>
> > **All remaining requested changes have already been addressed in the Weaknesses section. We thank again the reviewer for the helpful comments and for the thoughtful evaluation of our submission.**

---

> > ### Comment · Reviewer_nTQe · 2025-12-26
> >
> > Thank you for addressing the concerns and carrying out the additional empirical evaluations. For the SAITS hyper-parameter tuning, what was train-val-test split used for all the datasets? It would be useful to add this detail

---

> ### Author Response · Authors · 2025-12-26
> **Answer**
>
> We thank the reviewer for the positive feedback. We also appreciate this valuable comment and will clarify this point in Appendix F. For the hyperparameter search of SAITS, the train/validation/test split follows a 0.7/0.1/0.2 ratio along the temporal axis, consistent with all other univariate experiments in the paper (see Appendix B.1).

---

### Review · Reviewer_A38v · 2025-11-27

**Summary Of Contributions:**

The paper presents an analysis of two models, TabPFN-TS and MOTM, for time-series imputation. The authors evaluate these models on 33 out-of-distribution (OOD) datasets with 4 different settings. The paper is empirical and provides multiple scenarios with TabPFN-TS and MOTM and the comparison with existing supervised baselines.



Strengths:

- The biggest strength of the paper is that it is thoroughly evaluated on multiple experimental setups as well as several baselines.
If the question is between using TabPFN and MOTM for time series imputation then the work provides the answer for that

Weakness:

- The authors refer to time-indexed models, but I was not able to find a clear definition or reference for this specific term. I recommend defining it early in the paper, as its usage without explanation may be confusing to readers who are unfamiliar with it.

- While the paper reports some interesting comparative results between the two models, it would benefit from a deeper inspection of why these performance differences arise. Additionally, providing more actionable guidance for practitioners, such as scenarios where one model is preferable, failure modes,would substantially improve the paper’s practical impact.
- Although the two models evaluated are established, using only two methods may be insufficient for a purely empirical study. The authors could consider incorporating additional recent models such as TabICL and explore if it can they handle tabular/time-series structures in a manner similar to TabPFN to strengthen the empirical comparison.
- The writing is confusing especially in result interpretation section, some examples:
1. “The CD diagram further confirms that these two models are statistically superior compared to all others, except MoTM compared to SAITS.”: Then the statement should be TabPFN is superitor and MoTM is not.
2. “Local baselines remain resilient. Classical approaches leveraging temporal priors still deliver reasonable performance…”
 It would be helpful to specify what is meant by reasonable performance, perhaps by grounding the statement in concrete metrics or observations.
3. “TabPFN-TS consistently achieves the lowest MAE but at the cost of substantially higher inference time…”
 This wording suggests a trade-off between performance and inference time. If such a trade-off is intended or empirically supported, it should be articulated more explicitly; otherwise, the relationship should be clarified.



Relevant works not mentioned and cited:
1. Explore the Time Series Forecasting Potential of TabPFN Leveraging the Intrinsic Periodicity of Data - FMSD @ ICML 2025
2. ImputeFormer: Low Rankness-Induced Transformers for Generalizable Spatiotemporal Imputation, KDD 2024
3. ImputeINR: Time Series Imputation via Implicit Neural Representations for Disease Diagnosis with Missing Data, IJCAI 2025

**Audience:**

Yes

**Audience Explanation:**

This work can be interesting for the Time series imputation community.

**Claims And Evidence:**

Yes

**Claims Explanation:**

The major claim of the paper in conclusion is TabPFN-TS is better than any of the existing approaches for time series imputation, and the work empirically shows that.

**Requested Changes:**

* Please make the sentences in the result interpretation backed by data, as pointed out in weakness
* Can you present an investigation on why TabPFN is so good at this particular task?
* Add other foundation models as well, as mentioned in the weakness and maybe also explore if other tabular foundation models can be directly used for time series imputation such as tabpfn.
* Provide more insights like limitations and bottlenecks
* Rewrite the titles of subsections, make them more specific; titles like Main results, are not very helpful for the reader.

---

> ### Author Response · Authors · 2025-12-11
> **(Answer 1/2)**
>
> We sincerely thank the reviewer for taking the time to provide this detailed feedback.
>
> ## Weaknesses
>
> > **Weakness 1.** The authors refer to time-indexed models, but I was not able to find a clear definition or reference for this specific term. I recommend defining it early in the paper, as its usage without explanation may be confusing to readers who are unfamiliar with it.
>
>  **Answer weakness 1.**  Thanks for pointing out this lack of clarity. It is true that time-index modelling is a less-studied approach compared to the mainstream patch-based or history-based time series models. We kindly refer to Appendix C in (Woo, 2023 [1]) for an in-depth discussion of deep time-index models. In addition, we updated our manuscript by citing this reference in the Introduction.
>
> [1] Woo et al., *Learning deep time-index models for time series forecasting*, ICML 2023
>
> > **Weakness 2.** While the paper reports some interesting comparative results between the two models, it would benefit from a deeper inspection of why these performance differences arise. Additionally, providing more actionable guidance for practitioners, such as scenarios where one model is preferable, failure modes,would substantially improve the paper’s practical impact.
>
>  **Answer weakness 2.**  Thank you for this suggestion. Empirically, TabPFN-TS outperforms MoTM on almost all datasets we tested, leaving few dataset-dependent cases where MoTM is preferable in terms of accuracy. The main practical distinction therefore lies in computational constraints: TabPFN-TS can become computational-intensive for long sequences or many covariates, whereas MoTM scales more smoothly.
>
> Consequently, the most actionable guidance is hardware-driven. Use TabPFN-TS whenever GPUs are available and prefer MoTM in settings with resource constraints or when the number of sequences to predict becomes the bottleneck. Appendix D.2 now documents these trade-offs and failure modes.
>
> > **Weakness 3.** Although the two models evaluated are established, using only two methods may be insufficient for a purely empirical study. The authors could consider incorporating additional recent models such as TabICL and explore if it can they handle tabular/time-series structures in a manner similar to TabPFN to strengthen the empirical comparison.
>
>  **Answer weakness 3.** Ensuring a thorough empirical evaluation of the time-indexed foundation models is indeed an important point. While we selected diverse sets of representative task-specific and local baselines, no other foundation model was included in our main experiments. Below we aim to provide some elements to motivate this choice.
>
> In Appendix A.4, we discussed why MOMENT, another popular Tranformer-based foundation model for time series imputation, was excluded from the main experiments.
> Following on another request from reviewer nTQe, we updated Appendix A.4 with additional experiments assessing the zero-shot imputation performances of MOMENT as well as NuwaTS, another Transformer-based imputation foundation model.
> These experiments carried out on 11 datasets unambiguously demonstrate that both MOMENT and NuwaTS lag significantly behind TabPFN-TS or MoTM.
> Consequently, they are not included in the main benchmark.
>
>
> Finally, we agree that looking for other zero-shot regressors is a promising avenue of research.
> However, building tabular foundation models for regression is a recent field and, to the best of our knowledge, no alternative to TabPFN emerged at the time of writing.
> Particularly, TabICL seeks to perform faster inference than TabPFN but was designed and evaluated only on classification tasks.
> As such, it is not applicable to our regression setting.
>
> > **Weakness 4.** The writing is confusing especially in result interpretation section, some examples (...).
>
>  **Answer weakness 4.** Thank you for your careful reading of the paper, we updated it following your advice to avoid confusion.
> In particular, the third example you mention (our comment about TabPFN's inference time) is indeed a wording issue, as there is no explicit trade-off between performance and inference time for TabPFNv2.
> It is worth mentioning that the inference time of TabPFNv2 is a known issue, which has motivated further research, e.g. TabICL, a foundation model for tabular classification problems).
>
> > **Weakness 5.** Relevant works not mentioned and cited ...
>
> Thank you for these valuable references. We have added them to the paper.

---

> > ### Comment · Reviewer_A38v · 2025-12-16
> > **Additional Comments**
> >
> > > Consequently, the most actionable guidance is hardware-driven. Use TabPFN-TS whenever GPUs are available and prefer MoTM in settings with resource constraints or when the number of sequences to predict becomes the bottleneck. Appendix D.2 now documents these trade-offs and failure modes.
> >
> > Considering that this is key guidance, I do not think it belongs to appendix here. I also see that the discussion around why TabPFN is better, and the author's answer is:
> >
> > > these results underline the importance of well-designed time features and suggest that part of
> > TabPFN-TS’s effectiveness stems from the alignment between its meta-training distribution and the provided
> > temporal encodings.
> >
> > is in the appendix. It is an interesting finding for the reader. I think this should be included in the main text.
> >
> > The sentence
> >
> > > while MoTM ranks second (MAE = 0.371, avg. rank = 3.62), statistically on a par with SAITS
> >
> > Is incorrect, MoTM Ranks 3rd. According to the CD diagram(and second in the figure above, but it's hard to guess anything without raw results), can the authors also guide me on where I can find the raw results for section 3.1.1
> >
> > Also, the paper cited for Critical difference is a survey paper for time series classification; it's more apt to cite the original paper[1], while you can add a footnote that you used the code from another paper for the CD diagram:
> >
> > [1] Statistical Comparisons of Classifiers over Multiple Data Sets, Janez Demsar, JMLR
> >
> > Overall, the rebuttal answers some of my questions, but the writing should be improved, and the authors should select actionable insights from additional experiments and put that in the main text.

---

> > > ### Author Response · Authors · 2025-12-16
> > > **Response to the Reviewer’s Additional Comments**
> > >
> > > Thank you for the additional clarifications and suggestions.
> > >
> > > We agree with the reviewer that two key actionable takeaways from our additional analyses: (i) compute-driven deployment guidance and (ii) the role of input feature design in TabPFN-TS—should be visible in the main paper rather than confined to the appendix. In the revised manuscript, we therefore (a) add a short “Practical deployment considerations.” paragraph in Section 4.1, summarizing when to prefer TabPFN-TS versus MoTM, with a pointer to Appendix D.2 for full details, and (b) add a short paragraph in Section 3.1 highlighting the impact of temporally encoded features based on the ablation results (Appendix D.1), while keeping the full experimental breakdown in the appendix. These points are also reflected in the Conclusion, where we outline as future work how to automatically identify effective input feature encodings for TabPFN-TS and, more broadly, how to design stronger time-indexed foundation models, which is the overarching goal of this paper.
> > >
> > > Regarding the ranking statement, we thank the reviewer for pointing out this error. The reviewer is correct that MoTM ranks 3rd by average rank in the CD analysis. The confusion arose because MoTM is 2nd in aggregated mean MAE in our summary table, while ranking 3rd in average rank across datasets. We correct the sentence in Section 3.1.1 accordingly, and we explicitly state that MoTM and SAITS are not significantly different under the CD/Nemenyi comparison (as indicated by the CD diagram), which was the intended point.
> > >
> > > Concerning the raw results for Section 3.1.1, we agree with the reviewer that the bar plot and CD diagram alone are insufficient for verification. In the revision, we add an explicit pointer to the full per-dataset results reported in Table 6 (Appendix C.1).
> > >
> > > Finally, we thank the reviewer for the citation correction: we now cite the original reference for statistical comparisons and critical difference diagrams, *Demsar, “Statistical Comparisons of Classifiers over Multiple Data Sets”, JMLR.*
> > >
> > > To summarize, we acknowledge the reviewer’s point that the writing and result interpretation should be tightened. We therefore revised Sections 3.1.1 and Section 4 for clarity and correctness, and moved the main actionable points into the main text. All the changes are marked in **orange** in the revised version of our manuscript.

---

> ### Author Response · Authors · 2025-12-11
> **(Answer 2/2)**
>
> ## Requested Changes
>
> > **Requested change 2.** Can you present an investigation on why TabPFN is so good at this particular task?
>
> **Answer requested change 2.** Thank you for this very relevant question. We now include in Appendix D.1 an ablation study specifically analyzing why TabPFN-TS performs so well on this task. In this analysis, we systematically vary the temporal encodings used as input and evaluate the impact across 11 representative datasets. The results show that the original design (daily + weekly Fourier features, complemented by the raw time index) consistently provides the best zero-shot performance. Removing informative frequencies or replacing them with random ones leads to a clear drop in accuracy, whereas adding spurious frequencies has only minor effects.
>
> Overall, these findings highlight the importance of well-constructed time features and suggest that part of TabPFN-TS’s effectiveness comes from the strong alignment between the prior distribution of TabPFN and the temporal encodings supplied at inference. More broadly, this aligns with the intuition that in-context learning is particularly effective when the input context densely covers the structure of the underlying domain; here, the normalized temporal grid supported by the Fourier features.
>
> > **Requested change 4.** Provide more insights like limitations and bottlenecks
>
> **Answer requested change 4.** As specified above, the newly added Appendix D.1 investigates a key limitation of TabPFN-TS, its dependence on specific time features manually designed.
> This investigation takes the form of an ablation study that clearly confirms the importance of well-designed time features.
> We conclude this section by discussing future work on the automatic extraction or selection of time features.
> Besides, we included a discussion of the computational bottlenecks in Appendix D.2.
>
> > **Requested change 5.**  Rewrite the titles of subsections, make them more specific; titles like Main results, are not very helpful for the reader.
>
> **Answer requested change 5.** Thanks, we reformulated the titles accordingly.
>
>
>
> > **All of the remaining requested changes have been addressed in the "Weaknesses" section. We would like to thank the reviewer again for the very helpful comments and for taking the time to review our work.**

---

### Author Response · Authors · 2025-12-11
**General Comment**

We thank all reviewers for their constructive feedback and suggestions. We addressed each comment carefully and revised the paper accordingly. Following the reviewers’ recommendations, we also added new experimental results, in particular regarding :

- the sensitivity of TabPFN-TS to temporal input features (Appendix D.1, Reviewers A38V, E9qb),
- a discussion on the integration of non-aligned covariates (Appendix E.2; Reviewers A38V, nTQe),
- improved descriptions of the supervised baselines and hyperparameters search for SAITS (Appendix A.2 and Appendix F; Reviewers ixDG, nTQe),
- quantitative results and insights explaining why baselines such as MOMENT and NuwaTS were not included in the main paper (Appendix A.4; Reviewer nTQe).
- the code has been shared via an anonymous link: https://anonymous.4open.science/r/tmlr-submission-timeindexed-tsfm/. (Reviewer ixDG)

All modifications and newly reported results **are highlighted in blue** in the **revised version of the manuscript**.

---

### Decision · Action_Editor_jPAZ · 2026-01-07

**Recommendation:** Accept as is

**Audience:**

Yes

**Audience Explanation:**

All reviewers indicate that the work would be of interest to at least part of the TMLR audience, particularly researchers and practitioners working on time-series imputation and foundation models. Several reviewers explicitly note the community value of the large-scale benchmark.

**Claims And Evidence:**

Yes

**Claims Explanation:**

All reviewers indicate that the main claims are supported by the empirical results. One reviewer (ixDg) initially raised concerns about training procedures and reproducibility, but later confirmed that the revisions and released code made the evidence more convincing.